# Quantifying Structure in CLIP Embeddings:
# A Statistical Framework for Concept Interpretation

**Jitian Zhao**                                                                     *jzhao326@wisc.edu*
*Department of Statistics, University of Wisconsin-Madison*

**Chenghui Li**                                                                      *cli539@wisc.edu*
*Department of Statistics, University of Wisconsin-Madison*

**Frederic Sala**                                                                   *fredsala@wisc.edu*
*Department of Computer Science, University of Wisconsin-Madison*

**Karl Rohe**                                                                       *karl.rohe@wisc.edu*
*Department of Statistics, University of Wisconsin-Madison*

**Reviewed on OpenReview:** *https://openreview.net/pdf?id=D6KOWi3kRY*

## Abstract

Concept-based approaches, which aim to identify human-understandable concepts within a model's internal representations, are promising for interpreting embeddings from deep neural network models, such as CLIP. While these approaches help explain model behavior, current methods lack statistical rigor, making it challenging to validate identified concepts and compare different techniques. To address this challenge, we introduce a hypothesis testing framework that quantifies rotation-sensitive structures within the CLIP embedding space. Once such structures are identified, we propose a post-hoc concept decomposition method. Unlike existing approaches, it offers theoretical guarantees that discovered concepts represent robust, reproducible patterns (rather than method-specific artifacts) and outperforms other techniques in terms of reconstruction error. Empirically, we demonstrate that our concept-based decomposition algorithm effectively balances reconstruction accuracy with concept interpretability and helps mitigate spurious cues in data. Applied to a popular spurious correlation dataset, our method yields a 22.6% increase in worst-group accuracy after removing spurious background concepts.

## 1 Introduction

CLIP has become a foundational model for a wide range of visual applications (Radford et al., 2021). Interpreting its high-dimensional embeddings is challenging due to the complex and entangled nature of the learned representations. Recent works address this via *concept-based decomposition*, aiming to identify interpretable semantic patterns within model components, embeddings, and neurons (Gandelsman et al., 2024a;b; Balasubramanian et al., 2024).

Existing approaches broadly fall into two categories. The first category, exemplified by SpLiCE (Bhalla et al., 2024), decomposes CLIP embeddings into sparse, human-interpretable concepts such as words. While this sparse decomposition improves interpretability, it introduces reconstruction errors, meaning that a substantial portion of the original embedding's information is lost. This negatively impacts downstream tasks such as zero-shot classification, where preserving semantic information is crucial. The second category uses Singular Value Decomposition (SVD) to decompose embeddings into linear combinations of concept vectors (Fel et al., 2024; Graziani et al., 2023; Zhang et al., 2021). These methods maintain high reconstruction fidelity (i.e., the reconstructed embedding closely approximates the original with minimal error by filtering

out noise). However, they often struggle with concept interpretability, as singular vector directions are not inherently aligned with human-interpretable concepts, making their meaning largely dependent on human intuition. This highlights a trade-off: methods that prioritize interpretability often sacrifice reconstruction fidelity, while those that preserve fidelity tend to lack meaningful concept alignment.

Beyond this seeming interpretability-reconstruction fidelity trade-off, ***an even deeper issue remains***: existing methods are ad-hoc rather than the result of a rigorous statistical framework. In real-world settings, embeddings are inherently noisy, and without statistical guarantees, it is unclear whether the concepts extracted by a given method capture meaningful structure or merely reflect artifacts of noise. A key challenge is that noise and meaningful structure can both produce seemingly interpretable components, leading to silent failures where methods extract spurious "concepts" that do not correspond to real semantic attributes.

To address both challenges, we first propose a hypothesis testing framework to detect rotation-sensitive structure in the embedding subspace. Our key insight is that semantic concepts manifest as directional patterns in embedding space that are *sensitive to rotation, unlike random noise*, which remains statistically unchanged under rotation. By testing for rotation-sensitive patterns, our method distinguishes noise from true underlying structure, ensuring that extracted concepts reflect meaningful, stable properties of the data rather than arbitrary artifacts.

Table 1: Sparsity and Fidelity Across Techniques

|          | SVD | SPLICE | Ours |
|----------|-----|--------|------|
| **Sparsity** |     | ✓      | ✓    |
| **Fidelity** | ✓   |        | ✓    |

Building on the theoretical insights of Varimax rotation (Rohe & Zeng, 2020), we develop a post-hoc method that requires no additional training or human annotation. Our approach ***achieves both the interpretability benefits of sparse decomposition and the high reconstruction fidelity of SVD-based methods while offering statistical guarantees of concept recoverability***.

The remainder of this paper is organized as follows:

- In Section 3, we present a hypothesis testing framework to quantify rotation-sensitive concept structure in the embedding space. We provide the detailed test procedure, theoretical guarantees for test statistics, as well as empirical results.
- In Section 4, we introduce a post-hoc concept-based decomposition method accompanied with an automatic concept interpretation algorithm.
- In Section 5, we formalize a statistical model that connects concepts with embeddings and prove concept identifiability under certain assumptions. We show that concept decomposition methods with a fixed, misspecified concept vocabulary can suffer from reconstruction errors.
- In Section 6, we show through qualitative analysis that our method learns interpretable concepts and maintains high reconstruction fidelity, as evidenced by a sparsity-performance trade-off analysis. We also show that our method is effective in identifying and removing spurious concepts.

## 2 Background and Overview

We start with a brief discussion on statistical tests for rotation-sensitive structure and their implications for interpretability in neural embeddings.

**Hypothesis Test for Rotation-Sensitive Structure** Early multivariate inference framed "no preferred direction" as a null hypothesis, leading to tests for departures from *spherical symmetry*. Classical methods such as Mauchly's test for sphericity (Mauchly, 1940), John's test for identity covariance (John, 1971), and the Bingham test (Bingham, 1974), all treat rotational invariance as the baseline, so significant rejections indicate directional (rotation-sensitive) structure. In high dimensions, random-matrix approaches extend this logic by comparing observed eigen-spectra to isotropic nulls (Ledoit & Wolf, 2002; Chen et al., 2010). Building on these ideas, we adapt rotation-invariant bootstrap tests to neural embeddings, where coordinates are arbitrarily scaled or rotated. This allows us to assess deviations from isotropy before applying Varimax for interpretability.

**Rotation-sensitive Structure and Factor Interpretability** In classical factor analysis, raw factors are *rotation-indeterminate*: any orthogonal transformation of the loading matrix yields the same likelihood, so interpretability depends on selecting a "simple-structure" rotation such as Varimax (Kaiser, 1958). Recent work shows Varimax is not merely cosmetic—it can consistently recover sparse structure under mild conditions (Rohe & Zeng, 2023). Independent component analysis (ICA) addresses the same indeterminacy by leveraging non-Gaussianity to make the rotation identifiable, illustrating how probabilistic assumptions can anchor axes meaningfully (Hyvärinen et al., 2023). In modern representation learning, this symmetry persists: embeddings from contrastive or language-model objectives are only identifiable up to an unknown linear map, making axis-aligned interpretations fragile unless the rotation is fixed (Roeder et al., 2021). Empirically, post-hoc rotations can sharpen semantics; for example, rotating word embeddings can align dimensions with human concepts without degrading performance (Park et al., 2017).

## 3 Hypothesis Test for Rotation Sensitive Concepts

We develop a hypothesis testing framework to detect concepts in embedding spaces through rotational properties. We first characterize meaningful concepts through rotational sensitivity (Sec. 3.1), validating our intuition on synthetic and real datasets. We then develop a resampling procedure (Sec. 3.2), test statistics (Sec. 3.3), and test procedure with experimental results in (Sec. 3.4).

### 3.1 Characterizing Meaningful Structure via Rotational Sensitivity

We first explain why a rotation-based approach is well-suited for statistically modeling concepts in embeddings. Neural networks process embeddings through inner products with weight vectors, which measure how closely the embedding aligns with each weight vector's direction, making embeddings inherently directional objects. When examining embedding spaces for meaningful structure, we essentially ask whether the distribution of embeddings shows preferences for certain directions over others. A completely structureless embedding space samples points uniformly from all directions, making it *rotationally invariant*. In contrast, meaningful concepts manifest as preferred directions in the embedding distribution, breaking this invariance.

In the case of CLIP, this directional structure is not merely empirical but is enforced by the training objective itself. CLIP's contrastive loss maximizes the dot product similarity between matched image and text embeddings (Radford et al., 2021), which imposes a *linear readout* property on the final representation layer: for the model to correctly associate a concept (e.g., "dog"), the image embedding must have a high projection onto the corresponding direction in text embedding space. Unlike intermediate layers, where semantic content may be encoded nonlinearly, the final CLIP embedding space must therefore be linearly decodable by design, making singular vectors and linear decomposition the appropriate analytical tools for this space.

To accurately characterize these rotational preferences, we focus on the rotational properties of singular vectors rather than raw embeddings. This choice is crucial because heterogeneous scaling in different directions (i.e., elliptical patterns in data) can create apparent rotation sensitivity in the raw embeddings even when no meaningful structure exists. Singular vectors, being normalized to unit variance, allow us to identify true directional preferences while controlling for such scaling effects.

We now formalize these ideas, starting with a formal definition of rotational invariance:

**Definition 1** (Probabilistic Rotational Invariance). *A probability distribution with density function $f$ on $\mathbb{R}^d$ is said to be* rotationally invariant *if for any rotation matrix $R \in \mathbb{R}^{d \times d}$ (i.e., $R^\top R = I_d$ and $\det(R) = 1$), the distribution of $x$ is the same as the distribution of $xR$, $f(x) = f(xR)$ for all $x \in \mathbb{R}^d$ and all rotation matrices $R$.*

Intuitively, this definition formalizes when a distribution looks the same in all directions; there are no preferred directions or patterns in the data. To illustrate this definition, we first examine classical examples of rotation invariance, both at the data distribution level and the concept level:

**Example 1** (Standard Multivariate Normal is Rotationally Invariant). *The multivariate Gaussian distribution $\mathcal{N}(0, I_d)$ is rotationally invariant.*

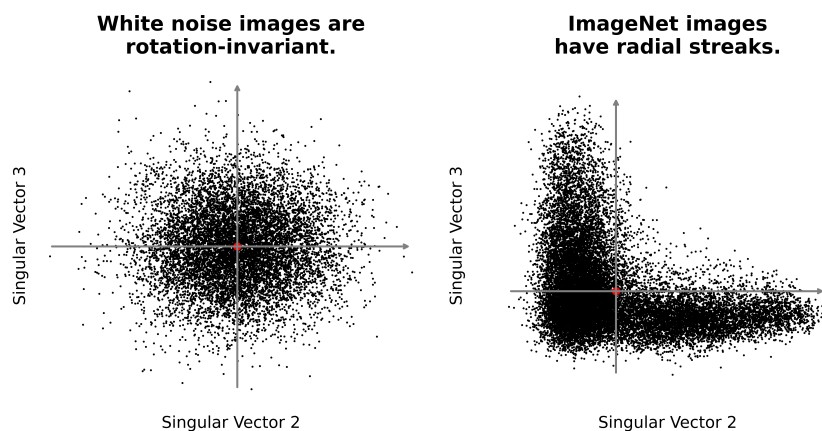

Figure 1: Visualization of singular vector loadings from CLIP embeddings from the ViT-B/32 backbone model, where loadings represent how much each singular vector contributes to an image's representation. Left: Projection onto 2nd and 3rd singular vectors (after Varimax rotation) of embeddings from white noise images, showing rotation-invariant structure. Right: Same projection for ImageNet validation images, revealing distinct radial streak patterns that indicate rotation-sensitive structure. Each point represents one image, with the first singular vector excluded to remove mean effects.

While Example 1 demonstrates rotation invariance at the data level, we are particularly interested in the rotational properties of concept-specific patterns, which we capture through singular vectors:

**Example 2** (Singular Vectors of Gaussian Noise are Rotationally Invariant). *Let $A \in \mathbb{R}^{n \times d}$ be a matrix with entries $A_{ij}$ drawn i.i.d. from $\mathcal{N}(0,1)$. Then the first $k$ left singular vectors $U \in \mathbb{R}^{n \times k}$ and right singular vectors $V \in \mathbb{R}^{d \times k}$ from the truncated SVD of $A$ are both rotationally invariant.*

This serves as our null model, representing the absence of meaningful concept structure. In contrast, embeddings that encode meaningful concepts exhibit rotation sensitivity. For example,

**Example 3** (Gaussian Mixture Model Shows Rotation Sensitivity). *Consider data drawn from a Gaussian mixture model: $x \sim \frac{1}{2}\mathcal{N}(\mu, I_d) + \frac{1}{2}\mathcal{N}(-\mu, I_d)$, where $\mu = (1, 0, 0, \dots, 0)^\top \in \mathbb{R}^d$. This distribution is not rotationally invariant.*

We can further validate these examples using real CLIP embeddings in Figure 1. The left panel shows embeddings of white noise images, displaying uniform distribution from all directions (i.e., rotation invariance) similar to Example 2. The right panel shows ImageNet embeddings, revealing clear directional structure through radial streaks, analogous to the structured distribution in Example 3.

To formalize these ideas, let $A \in \mathbb{R}^{n \times d}$ be the data matrix. We obtain its truncated singular value decomposition, where $U \in \mathbb{R}^{n \times k}$ contains the first $k$ left singular vectors of $A$. Each column of $U$ represents a principal direction of variation in the data. Our hypothesis test specifically examines the rotational properties of left singular vectors (matrix $U$).

## 3.2   Sampling from the Rotation invariant Distribution

Rotation sensitivity only manifests when an embedding prefers certain directions. To mimic the absence of such structure, we generate a null sample by *independently* rotating each row of the embedding while preserving its length. We describe the details of this method in Algorithm 4, which is deferred to the Appendix. Comparing any test statistic to the distribution obtained from these rotated replicas yields a Monte-Carlo $p$-value for the presence of rotation-sensitive patterns.

The following proposition establishes the theoretical guarantees for our resampling procedure:

**Proposition 1** (Statistical Properties of Resampling). *Let $\{x_i\}_{i=1}^n \subset \mathbb{R}^d$ be i.i.d. samples from a probability distribution with density $f$. For any measurable test statistic $T : \mathbb{R}^{d \times n} \to \mathbb{R}$, define:*

$$T_1 = T(x_1, \ldots, x_n), \quad x_i^{rot} = R^i x_i, \quad R^i \overset{iid}{\sim} Uniform(SO_d), \quad T^* = T(x_1^{rot}, \ldots, x_n^{rot}).$$

*If $f$ is rotationally invariant, then $T^*$ and $T_1$ have the same distribution and are conditionally independent given the set of norms $\{\|x_i\|_2\}_{i=1}^n$.*

This proposition guarantees that under the null hypothesis of rotational invariance, the resampled test statistic ($T^*$) follows the same distribution as the original statistic ($T_1$).

### 3.3 Test Statistics

To detect and quantify rotation-sensitive structure in embedding spaces, we propose two complementary test statistics that capture different aspects of rotation sensitivity: distributional non-Gaussianity through kurtosis and achievable sparsity under rotation through the Varimax objective function.

**Kurtosis-based Statistic.** Our first test statistic measures the non-Gaussian patterns in $U$, as meaningful concepts typically deviate from normal distributions. We define:

$$TS_1(U) = \frac{1}{k} \sum_{i=1}^k |\text{kurtosis}(U_{\cdot i})|, \quad \text{where kurtosis}(X) = \frac{\mathbb{E}[(X - \mu)^4]}{(\mathbb{E}[(X - \mu)^2])^2} - 3,$$

with $\mu = \mathbb{E}[X]$. Under the rotation invariance null hypothesis, we expect this statistic to be close to zero.

**Theorem 1.** *Under the null model of Example 2, an equivalent rescaled version of $TS_1(U)$ follows a standard normal distribution.*

This normalization provides an efficient computational path for hypothesis testing under Gaussian assumptions. We defer the detailed proof to the Appendix.

**Varimax-based Statistic.** Our second test statistic optimizes over rotations to detect patterns that may be hidden in the original coordinate system. We define:

$$v(U, R) = \sum_{\ell=1}^k \frac{1}{n} \sum_{i=1}^n \left( |UR_{i\ell}|^4 - \left( \frac{1}{n} \sum_{q=1}^n |UR_{q\ell}|^2 \right)^2 \right), \qquad TS_2(U) = \max_{R \in SO_k} v(U, R), \tag{1}$$

where $v$ is the Varimax objective function, and $SO_k$ is the special orthogonal group of $k \times k$ rotation matrices. This statistic measures the maximum achievable sparsity under rotation, making it particularly sensitive to structured patterns that may be hidden in the original coordinate system.

To apply these test statistics, we use a bootstrap approach. We first generate samples from the null distribution by applying random rotations to the original data matrix. We then compute both test statistics on these null samples to form their empirical distributions. The p-values are calculated by comparing the observed test statistics against these null distributions.

### 3.4 Test Procedure and Results

We present a hypothesis testing procedure for detecting rotation-sensitive structures in embedding spaces. Under the null hypothesis of rotation invariance, we expect large p-values indicating no meaningful structure (e.g., $\geq 0.05$), while significantly small p-values (e.g., $\leq 0.05$) suggest the presence of rotation-sensitive structure. Algorithm 1 outlines our testing approach.

We evaluate our method using CLIP ViT-L/14 embeddings on three datasets: ImageNet validation set images, white noise images (Gaussian noise fed into the CLIP model), and pure white noise embeddings. As shown in Figure 6, our test statistics reveal strong evidence of rotation-sensitive structure in CLIP embeddings of ImageNet images, with both test statistics showing significant separation between their bootstrap

---

**Algorithm 1** Hypothesis Test for Rotation-Sensitive Concepts

---

**Require:** Matrix $U \in \mathbb{R}^{n \times k}$, resamples $N_{\text{resample}}$
**Ensure:** Varimax p-value $p_{\text{v}}$, kurtosis p-value $p_{\text{kur}}$

1: **for** $i = 1$ to $N_{\text{resample}}$ **do**
2:     $U_i^{\text{rot}} \leftarrow$ Rotation Invariant Matrix$(U)$        ▷ Algorithm 4
3:     $Z_i^{\text{rot}} \leftarrow U_i^{\text{rot}} \times \text{argmax}_{R \in \text{SO}_k} v(U_i^{\text{rot}}, R)$        ▷ Apply Varimax rotation in eq: equation 1
4:     Compute statistics $TS_1(Z_i^{\text{rot}})$, $TS_2(Z_i^{\text{rot}})$
5: **end for**
6: $\hat{Z} \leftarrow U \times \text{argmax}_{R \in \text{SO}_k} v(U, R)$
7: Compute $TS_1(\hat{Z})$, $TS_2(\hat{Z})$
8: Compute p-values:
9: $p_{\text{kur}} \leftarrow \dfrac{\sum \mathbb{1}[TS_1(\hat{Z}) > TS_1(Z_i^{\text{rot}})]}{N_{\text{resample}}}$
10: $p_{\text{v}} \leftarrow \dfrac{\sum \mathbb{1}[TS_2(\hat{Z}) > TS_2(Z_i^{\text{rot}})]}{N_{\text{resample}}}$
11: **return** $p_{\text{kur}}, p_{\text{v}}$

---

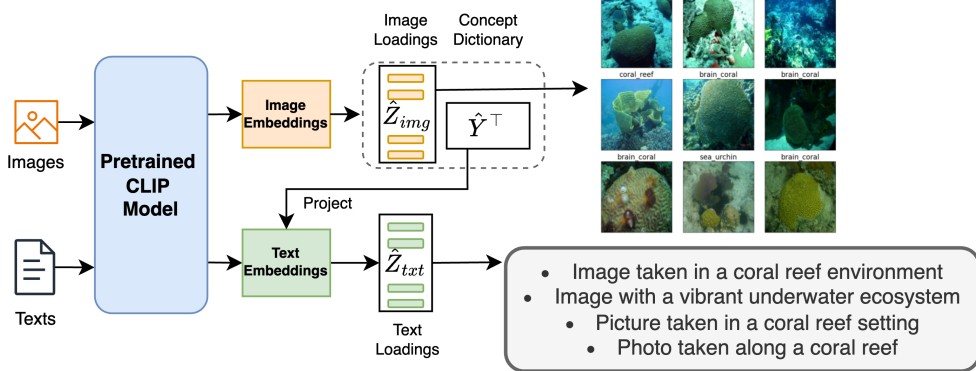

Figure 2: Pipeline overview. CLIP embeddings from images and texts are processed to extract interpretable concepts. Image embeddings are factorized into sparse loadings ($\hat{Z}_{img}$) and a concept dictionary ($\hat{Y}$), while text embeddings are projected to obtain loadings ($\hat{Z}_{txt}$). Bottom: Example of a discovered concept with its representative images and text descriptions.

null distributions and observed values. Control experiments in white noise *confirm our method's validity.* Both white noise embeddings (p-values: 0.55 for Kurtosis, 0.6 for Varimax) and white noise images (p-values: 0.62 for Kurtosis, 0.81 for Varimax) yield non-significant p-values, confirming that random noise contains no meaningful structure.

## 4    Identifying Interpretable Concepts

We present our main algorithm for identifying rotation-sensitive concept structure in the embedding space, as motivated by the notions in Section 3. Our approach decomposes the embedding matrix into two components: a sparse loading matrix that captures the relationships between individual images and concepts, and an orthogonal concept dictionary that maintains independence between the discovered concepts. This achieves both interpretability and high reconstruction fidelity.

### 4.1    Method Overview

As shown in Figure 2, our pipeline processes image and text inputs through a pretrained CLIP model to obtain embeddings. Given image embeddings $A \in \mathbb{R}^{n \times d}$ and text embeddings $T \in \mathbb{R}^{M \times d}$, we learn $k$ orthogonal concepts represented through three key matrices: concept dictionary matrix $\hat{Y}$, image loadings $\hat{Z}_{img}$, and

---

**Algorithm 2** Concept-based Embedding Decomposition

---

1: **Input:** Embedding matrix $A \in \mathbb{R}^{n \times d}$, number of concepts $k$
2: **Output:** Concept matrix $\hat{Y} \in \mathbb{R}^{d \times k}$, image loadings $\hat{Z} \in \mathbb{R}^{n \times k}$
3: $\tilde{A} \leftarrow \text{Normalization}(A)$
4: $U, D, V^\top \leftarrow \text{SVD}(\tilde{A})$
5: $R \leftarrow \text{argmax}_{R \in \text{SO}_k} v(UD, R)$            $\triangleright$ Optimizes objective in Eq. equation 1
6: $\hat{Z} \leftarrow UDR$
7: $\hat{Y} \leftarrow VR$
8: **return** $\hat{Z}, \hat{Y}$

---

text loadings $\hat{Z}_{txt}$. The image and text loadings indicate how strongly each image or text embedding aligns with the learned concept - higher values represent a stronger association with a particular concept in the dictionary.

Algorithm 2 details our decomposition method, which has three key steps. First, we normalize the embedding matrix for better spectral estimation (see details in Appendix B). Second, we perform SVD to identify the principal directions in the embedding space. However, these raw SVD components, while capturing the underlying structure, are not automatically aligned with human-interpretable concepts. This motivates our third step: Varimax rotation, which maximizes the variance of squared loadings for each concept, naturally pushing individual loadings toward either high values or zero and thus promoting sparsity. This sparsification is crucial for interpretability — it associates each data point primarily with its most relevant concepts and makes concepts more semantically distinct by connecting them only to related examples. The rotation step effectively aligns the rotation-sensitive structure we detected (Section 3) with interpretable axes in the embedding space. We empirically validate the necessity of this rotation in Section 6, where we show that rotated sparse concepts exhibit more explicit semantics compared to raw SVD components.

While raw directions can be polysemantic, Varimax rotation directly addresses this entanglement by seeking the simplest available linear basis. Our results on Waterbirds (Section 6.3) confirm that this linear disentanglement is sufficient to isolate and remove spurious concepts such as backgrounds, yielding substantial improvements in worst-group accuracy.

Our framework thus bridges rotation sensitivity and interpretability in two steps. First, the hypothesis test (Section 3) uses rotation sensitivity to distinguish signal from noise—isotropic distributions are rotation-invariant, so rotation sensitivity reveals "preferred directions" confirming non-random underlying structure. Second, Varimax rotation transforms these directions into sparse concept loadings, which is key to human interpretability: it associates data points only with their most relevant concepts and makes concepts semantically distinct. The connection between sparsity and interpretability is well-established in representation learning (Subramanian et al., 2018; Guillot et al., 2023; Faruqui et al., 2015).

## 4.2 Concept Interpretation

We propose two methods to interpret each concept from the decomposition. The first method utilizes the image loading matrix $\hat{Z}$ to identify representative examples for each concept. For the $j$-th concept, we examine its corresponding column $\hat{Z}_{.j}$ and select the $r$ images with the highest loading scores. These typically share common semantic features, allowing us to derive an interpretable theme for the concept.

Our second method (Alg. 3) provides automatic concept interpretation through text descriptions without human intervention. This approach requires a pool of text descriptions for the image dataset, which can be obtained through LLMs or visual-language models (e.g., LLaVA proposed by Liu et al. (2023)). The algorithm projects these text descriptions onto our learned concept space and identifies the most relevant descriptions for each concept. In our experiments, we use the curated text description set from Gandelsman et al. (2024b), which provides general descriptions of ImageNet classes.

---

**Algorithm 3** Automatic Concept Interpretation

---

1: **Input:** Concept matrix $\hat{Y} \in \mathbb{R}^{d \times k}$, text descriptions $\{M_i\}_{i=1}^{M}$, text embeddings $T \in \mathbb{R}^{M \times d}$, descriptions per concept $r$
2: **Output:** Text descriptions $\{\mathcal{T}_j\}_{j=1}^{k}$ for each concept
3: $L \leftarrow T\hat{Y}$                                ▷ Project text embeddings to concept space
4: **for** each concept $j = 1, \ldots, k$ **do**
5:     $\mathcal{I}_j \leftarrow$ indices of top $r$ values in $L_{.j}$
6:     $\mathcal{T}_j \leftarrow \{M_i : i \in \mathcal{I}_j\}$
7: **end for**
8: **return** $\{\mathcal{T}_j\}_{j=1}^{k}$

---

# 5 Theoretical Results: Identification and Recovery Bounds

In this section, we establish theoretical guarantees for our concept decomposition method. We first demonstrate that our method can reliably identify meaningful concepts under specific statistical assumptions, thereby extending previous work on Varimax rotation identification. We then analyze fundamental limitations of fixed-concept approaches, demonstrating why adaptive concept learning is necessary.

## 5.1 Concept Identification

Our identification guarantees are based on a key insight: when embedding data exhibits sufficient statistical structure, a unique rotation (up to permutation) exists that aligns with interpretable concepts. We formalize this through the following assumptions:

**Assumption 1** (The identification assumptions for Varimax). *The matrix $Z \in \mathbb{R}^{n \times k}$ satisfies the identification assumptions for Varimax if all of the following conditions hold on the rows $Z_i \in \mathbb{R}^k$ for $i = 1, \ldots, n$: (i) the vectors $Z_1, Z_2, \ldots, Z_n$ are i.i.d., (ii) each vector $Z_i$ has $k$ independent random variables (not necessarily identically distributed), (iii) the elements of $Z_i$ have kurtosis $\kappa \geq 3$.*

The independence conditions (i) and (ii) ensure structural consistency across samples, while (iii) requires sufficient non-Gaussianity in the data. We relax the equal variance assumption from Rohe & Zeng (2020), allowing different concepts to have different strengths of expression. This is crucial. In the vintage sparse-PCA model, the data admit the factorization $X = ZBY'$, where $Z, Y$ are Varimax-rotated eigenvectors, and $B$ is the diagonal matrix of eigenvalues, left and right multiplied by rotation matrices. Because B absorbs all scaling, Z and Y can be rescaled without losing orthogonality. Our model instead factors the data as $X = ZY'$ for clear interpretation purposes, where $Z$ is the data loading on each concept, and $Y$ is the concept dictionary. Hence, any attempt to transfer scale from Z to Y would break the orthogonality of $Y$, which makes the concept dictionary harder to interpret.

**Theorem 2** (Varimax rotation identification). *Suppose that $Z \in \mathbb{R}^{n \times k}$ satisfies Assumption 1. Define $\tilde{Z} = Z - \mathbb{E}(Z)$. For any rotation matrix $\tilde{R} \in \mathcal{O}(k)$,*

$$\arg \max_{R \in \mathcal{O}(k)} \mathbb{E}\left(v(R, Z\tilde{R}^\top)\right) = \{\tilde{R}P : P \in \mathcal{P}(k)\},$$

*where $\mathcal{P}(k) = \{P \in \mathcal{O}(k) : P_{ij} \in \{-1, 0, 1\}\}$, is the full set of matrices that allow for column reordering and sign changes, and $v$ is defined in equation equation 1.*

Under our assumptions, this shows the Varimax objective identifies the correct concept rotation up to permutation.

## 5.2 Reconstruction Error Bounds for Fixed-Concept Methods

When the concept matrix is fixed, reconstruction errors arise from potential misalignment between predefined concepts and the ground-truth concept structure. To formalize this limitation, we denote the ground-truth

latent concept matrix as $C^* \in \mathbb{R}^{d \times k}$ and the fixed concept matrix (such as in SpLiCE) as $C_W \in \mathbb{R}^{d \times m}$ where $k \leq d < m$. We assume $C_W$ may fail to capture some information present in $C^*$, which we quantify through the following condition: $\min_{P \in \mathbb{R}^{m \times k}} \|C_W P - C^*\|_F \geq \delta$, where $P$ is an arbitrary projection matrix, $\delta > 0$ represents the minimum possible misalignment between the fixed and true concepts, and $\|\cdot\|_F$ represents the Frobenius norm.

**Theorem 3** (Fixed concept-decomposition method reconstruction error lower bound). *Given the misspecification condition above, consider $A \in \mathbb{R}^{n \times d}$ such that $A = Z^* C^{*\top}$ with positive $k$-th singular value, i.e. $\sigma_k(Z^*) > 0$, then we have $\min_{Z \in \mathbb{R}^{n \times m}} \|A - Z C_W^\top\|_F \geq \sigma_k(Z^*)\delta$, where $\sigma_k(Z) = \sqrt{\sigma_k(Z^\top Z)}$ is the absolute $k$-th largest singular value of $Z$.*

This theorem quantifies the risk of fixing the concept decomposition matrix in SpLiCE: when the predefined concept vocabulary cannot be aligned with the true concepts, reconstruction error is unavoidable. Our proposed method avoids this limitation by learning concepts from the data.

# 6 Experiment Results

We evaluate our method through qualitative and quantitative analyses. We assess the interpretability of learned concepts via visualizations and textual alignment, analyze the trade-off between sparsity and reconstruction fidelity, and demonstrate the effectiveness of our method in removing spurious correlations across multiple datasets.

## 6.1 Qualitative Evaluation of Discovered Concepts

We evaluate the effectiveness of our concept decomposition method visually and textually .

**Setup.** We apply our method to CLIP ViT-B/32 embeddings of the ImageNet validation set, using the curated text description set from Gandelsman et al. (2024a) that provides class-specific descriptions generated via ChatGPT.

**Results.** Figure 3 compares concept clusters discovered by our Varimax-rotated decomposition (left column) against those from raw SVD (right column). For two representative concepts, we display the top nine images by loading score ($\hat{Z}_{.j}$) alongside their automatically retrieved text descriptions from Algorithm 3. The top row shows a concept manually selected to demonstrate the effectiveness of our method, while the bottom row concept shows a randomly selected concept from a pool of 50. ***Our method consistently yields semantically coherent clusters (e.g. butterfly feeding scenes, screws) with concise, focused descriptions.*** In contrast, raw SVD clusters mixed themes such as furniture and animals, screws and knitwears, accompanied with broader, less specific descriptions. These differences demonstrate that Varimax rotation effectively isolates meaningful concept directions in the CLIP embedding space, resulting in far more structured and interpretable concept representations than standard SVD decomposition.

## 6.2 Sparsity-performance Trade-off

We analyze how the number of concepts in our decomposition affects reconstruction fidelity, measured by the cosine similarity between original and reconstructed embeddings.

**Setup.** We evaluate our method on the ImageNet validation set using two CLIP models: ViT-B/32 (512 dimensions) and ViT-L/14 (768 dimensions). We compare our results against those of SpLiCE (Bhalla et al., 2024) as a baseline.

**Results.** Figure 4 shows that reconstruction fidelity improves with increasing number of concepts $k$ for both models. ViT-L/14 consistently shows lower cosine similarity compared to ViT-B/32 at equal $k$, reflecting the challenge of capturing its richer 768-dimensional embedding space with the same concept budget. Our method ***achieves substantially higher reconstruction fidelity*** compared to SpLiCE when using comparable numbers of concepts.

The quality of reconstruction is crucial as it indicates how well our decomposition preserves the semantic information encoded in the original embeddings. While concept decomposition inherently involves a trade-off

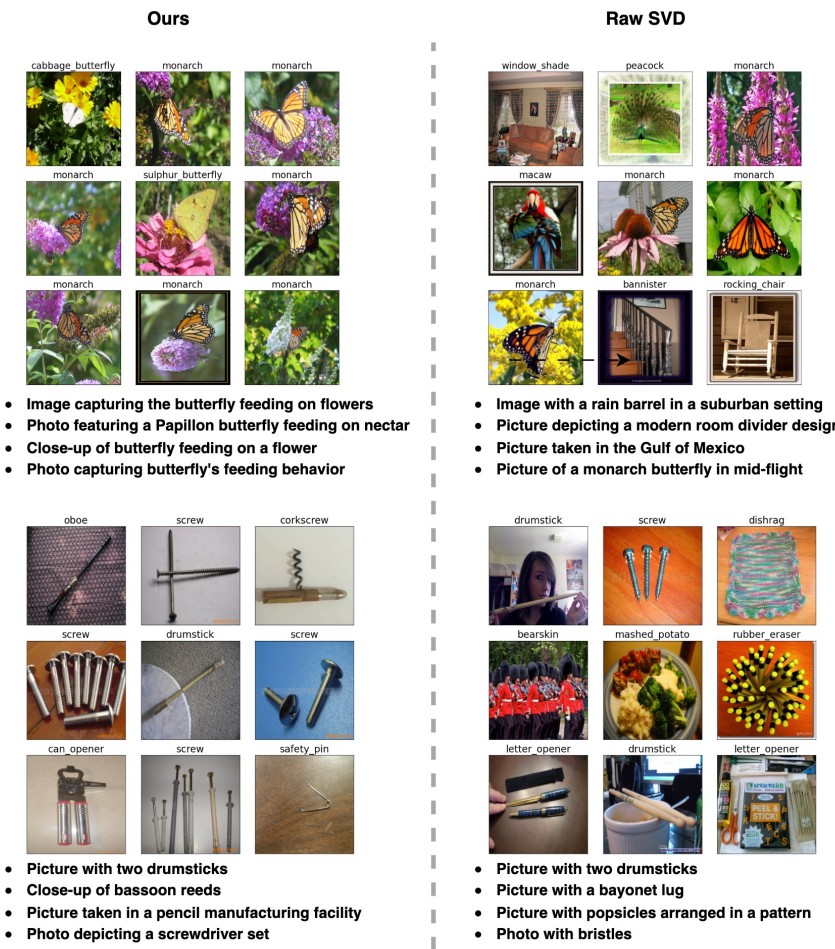

Figure 3: Comparison of concept clusters obtained by our Varimax-rotated decomposition (left column) and raw SVD (right column) on CLIP image embeddings. Each cell shows the top nine images for a given concept, annotated with their retrieved text descriptions. Our method produces tight, semantically coherent clusters and precise labels, whereas raw SVD yields mixed-semantics groups and more generic descriptions.

Table 2: Performance comparison across datasets. WG[†]: worst-group accuracy (higher better), Gap[‡]: accuracy gap (lower better), mF1: micro-F1, MRec: macro-recall. Best results in **bold**. Purple and green highlights indicate the best worst-group accuracy and the smallest accuracy gap.

| Model | Waterbirds | | | iWildCam | | | CelebA | | |
|---|---|---|---|---|---|---|---|---|---|
| | Avg | WG[†] | Gap[‡] | Acc | mF1 | MRec | Avg | WG[†] | Gap[‡] |
| ZS | 84.8 | 38.1 | 46.7 | 6.23 | 0.001 | 0.002 | 81.2 | 74.2 | 7.0 |
| SVD-recon. | 85.5 | 39.0 | 46.5 | 3.83 | 0.001 | 0.001 | 78.1 | 74.9 | **3.2** |
| Spurious Removed | **89.6** | **60.7** | **28.9** | **18.8** | **0.003** | **0.006** | **82.6** | **75.1** | 7.5 |

between interpretability and information preservation, our method offers flexible control through the number of concepts $k$, allowing users to balance these competing objectives. Unlike SpLiCE, which prioritizes interpretability at the cost of significant information loss, ***our approach maintains interpretable concepts while better preserving the original embedding structure***. This preservation of semantic information is essential for downstream applications and validates that our discovered concepts capture meaningful aspects of the data.

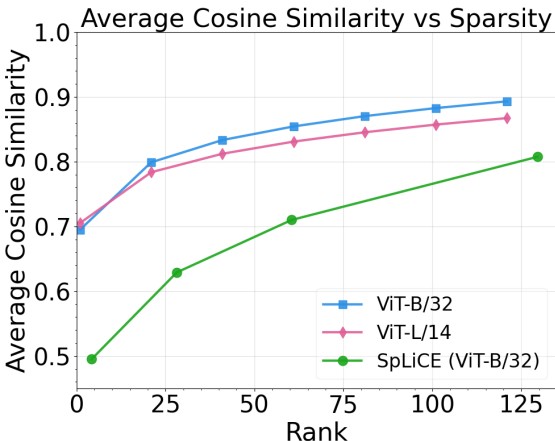

Figure 4: Reconstruction quality versus number of concepts. Higher cosine similarity indicates better preservation of the original embedding structure. Our method offers a better interpretability-fidelity trade-off than SpLiCE.

## 6.3 Removing Spurious Correlations

We evaluate our method's ability to identify and remove spurious correlations across three datasets: Waterbirds (Sagawa et al., 2019), WILDS-iWildCam (Beery et al., 2020), and CelebA (Liu et al., 2015). Dataset details are provided in Appendix B.

**Setup.** For all experiments, we use CLIP ViT-B/32 embeddings (512 dimensions) and decompose them into $k = 50$ concepts using Algorithm 2. Spurious concepts are identified using dataset-specific strategies (detailed in Appendix B), which analyze the correlation between concept embeddings and text descriptions emphasizing different attributes (e.g., target vs. background features). We defer the details of how to remove spurious concepts to the appendix.

We compare three embeddings: the full, original CLIP image embedding; the concept-based reconstruction, where embeddings are reconstructed from all learned concepts; and the spurious-removed reconstruction (Ours), where embeddings are reconstructed after removing spurious concepts. For classification, we follow the standard zero-shot classification setup.

**Results.** Our method ***consistently improves zero-shot prediction performance after removing spurious concepts*** (Table 2). On Waterbirds, removing spurious background concepts improves worst-group accuracy by 22.6% and reduces the accuracy gap by 17.8%. On iWildCam, the prediction accuracy triples from 6.23% to 18.8%; and on CelebA, we achieve the highest average and worst-group accuracy while using only 5% of the original embedding dimensions. The SVD-reconstructed embeddings maintain similar average accuracy to the original embeddings for the Waterbirds dataset, suggesting our method preserves task-relevant information while removing noise.

## 6.4 Additional Experiments

We present additional experiments in Appendix C. Specifically, Section C.4 evaluates our models across different backbones and datasets to demonstrate that our framework generalizes beyond CLIP. Section C.5 investigates model behavior when random rotations are applied to the CLIP embeddings. In Section C.6, we use a concept coherence metric to assess the semantic alignment of the discovered concepts. Finally, Section C.8 analyzes the alignment between our learned concept directions and the attention heads of the model's final layer.

# 7 Conclusion

We introduced a hypothesis testing framework to quantify rotation-sensitive structures in the embedding spaces and proposed a concept-decomposition method that achieves both high reconstruction fidelity and clear interpretability. We validated it through theoretical and empirical analyses. Applied to challenging distribution shift benchmarks, our method consistently demonstrated significant improvements after identifying and removing spurious concepts.

**Limitations.** Our approach assumes linearity in the embedding decomposition, which may overlook complex non-linear structures potentially present in the embedding space. Additionally, we note that the interpretability of discovered concepts depends partially on the quality and scope of the available text descriptions, which may introduce biases or limit generalization. Finally, while our hypothesis testing procedure is robust to rotationally invariant noise, it does not explicitly handle structured, non-rotational noise patterns, leaving room for further refinement in more nuanced settings.

**Acknowledgements.** The authors thank Kris Sankaran, Keith Levin, Yinqiu He, Changho Shin and Dyah Adila for the valuable discussions and feedback. Jitian Zhao and Chenghui Li gratefully acknowledge support from the IFDS at UW-Madison and NSF through TRIPODS grant 2023239 for their support.

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

Supplementary materials contain additional experiment details, results, and proofs. We provide glossary table in Section A, additional experiment details in Section B and results in Section C, technical lemmas and additional theoretical results in Section D, and proof in Section E.

## A  Glossary

The glossary is given in Table 3 below.

Table 3: Glossary of Notation

| Symbol | Definition |
|--------|------------|
| $U$ | Matrix containing singular vectors from SVD decomposition |
| $Z$ | Image loading matrix, represents how images load onto concepts |
| $\mathrm{SO}_k$ | Special orthogonal group (rotation matrices) in $\mathbb{R}^{k \times k}$ |
| $A$ | Input embedding matrix of size $\mathbb{R}^{n \times d}$ |
| $Y$ | Estimated concept matrix of size $\mathbb{R}^{d \times k}$ |
| $k$ | Number of concepts |
| $n$ | Number of data points/images |
| $d$ | Dimension of embeddings |
| $R$ | Rotation matrix |
| $T$ | Text embedding matrix |
| $C_j$ | The $j$-th concept |
| $\sigma_d(Z)$ | The absolute $d$-th largest singular value of $Z$ |
| $\boldsymbol{C}^*$ | Ground-truth latent concept matrix |
| $\boldsymbol{C}_W$ | Word concept matrix |
| $v(U, R)$ | Varimax objective function |
| $TS_1(U)$ | Kurtosis test statistic |
| $TS_2(U)$ | Varimax objective function test statistic |
| $TS_3(U)$ | Rescaled kurtosis test statistic |
| $\|\cdot\|_F$ | Frobenius norm |
| $\mathrm{kurtosis}(U_{.i})$ | Kurtosis of the $i$-th column of $U$ |
| $\mathcal{P}(k)$ | Set of permutation matrices in $\mathbb{R}^{k \times k}$ |

## B  Additional Experiment Details

All our experiment results are carried out using frozen pretrained weights from open-clip (ViT-B/32 and ViT-L/14), and no additional model training is involved.

### B.1  Concept Decomposition Algorithm Details

**Scaling Data Matrix**  We scaled the data matrix $A \in \mathbb{R}^{n \times d}$ before applying SVD in the concept decomposition algorithm. Define the row normalization vector as:

$$deg_r = A\mathbf{1}_d \in \mathbb{R}^n, \quad \tau_r = \frac{1}{n}\mathbf{1}_n^T deg_r \in \mathbb{R}, \quad D_r = \mathrm{diag}(deg_r + \tau_r \mathbf{1}_n) \in \mathbb{R}^{n \times n}$$

Similarly, define the column quantities $deg_c = \mathbf{1}_n^T A \in \mathbb{R}^d$, $\tau_c = \frac{1}{d} deg_c \mathbf{1}_d \in \mathbb{R}$, and $D_c = \mathrm{diag}(deg_c + \tau_c \mathbf{1}_d) \in \mathbb{R}^{d \times d}$. The scaled data matrix is then defined as $\tilde{A} = D_r^{-1/2} A D_c^{-1/2}$, which we use as input to the concept decomposition algorithm instead of the original matrix $A$. This scaling step with regularization parameters $\tau_r$ and $\tau_c$ helps stabilize the spectral estimation and prevents potential outliers in the singular vectors that could arise from noise in the data matrix (Le et al., 2017; Zhang & Rohe, 2018).

### B.2 Hypothesis Test Experiment Details

To validate our hypothesis testing framework, we conducted experiments on both a real dataset—the Im-ageNet validation set—and two synthetic datasets: white-noise image embeddings and pure white-noise embeddings. The goal was to assess the framework's ability to detect non-random structures in different types of data.

**Datasets.**

- **ImageNet Validation Set:** We used embeddings computed by a pretrained Vision Transformer (ViT-B/32) model on images from the ImageNet validation set.

- **White-Noise Image Embeddings:** We generated 10,000 white-noise images, each of size $224 \times 224$ pixels with 3 color channels. Each pixel value was drawn independently from a standard Gaussian distribution. These images were then processed by the pretrained ViT model to obtain embeddings of size $10,000 \times 512$.

- **Pure White-Noise Embeddings:** We directly generated a random noise matrix of dimensions $10,000 \times 512$, with each entry sampled from a standard Gaussian distribution, without passing through the embedding model.

**Experiment Details.** For each dataset, we obtained an embedding matrix $\tilde{A}$ as described above. We then performed Singular Value Decomposition (SVD) on $\tilde{A}$, decomposing it into $\tilde{A} = UDV^\top$. Here, $U$ is a matrix whose columns are the left singular vectors, representing orthogonal directions in the embedding space, and whose rows correspond to the images.

We observed that the first column of $U$ (the first principal component) often captured mean or bias effects in the embeddings. The loadings on this component were concentrated around a constant value, offering limited information about the latent structure of the data. Therefore, we excluded the first column of $U$, defining $\tilde{U} = U[:, 2:]$, to focus on more informative components.

Next, we applied our hypothesis testing framework to $\tilde{U}$ to compute p-values and test statistics, assessing the statistical significance of any non-random patterns present in the data.

**Randomness in the Procedure**

Our hypothesis testing procedure involves randomness in two key aspects:

1. **Row-wise Random Rotations:** To generate conditionally rotation-invariant data, we applied random rotations to each row of $\tilde{U}$. This step introduces randomness into the data transformation process.

2. **Generation of Synthetic Data:** The white-noise image and pure white-noise embeddings were generated using random sampling from standard Gaussian distributions.

To account for the variability introduced by these random processes, we performed additional tests using 5 different random seeds and varied the rank $k$ of $\tilde{U}$.

**Selection of Rank $k$** We investigated how the choice of rank $k$, the number of singular vectors retained in $\tilde{U}$, affects the results of the hypothesis tests. We expect the p-values to increase with larger $k$, indicating a decreased ability to detect rotation-sensitive structure. As $k$ increases, more columns of $U$ are included, potentially introducing additional noise and reducing the statistical power to detect non-Gaussian signals. We report our results in Figure 5. We observed that for white noise image embeddings, p-values increase as k increases, which aligns with our expectation. For white noise embedding, we observed p-values oscillate around 0.5 and show no clear pattern as $k$ changes. This aligns with our theoretical results from Example 2, which suggests $U$ follows a rotationally invariant distribution, and the p-value should follow an approximately uniform distribution between 0 and 1.

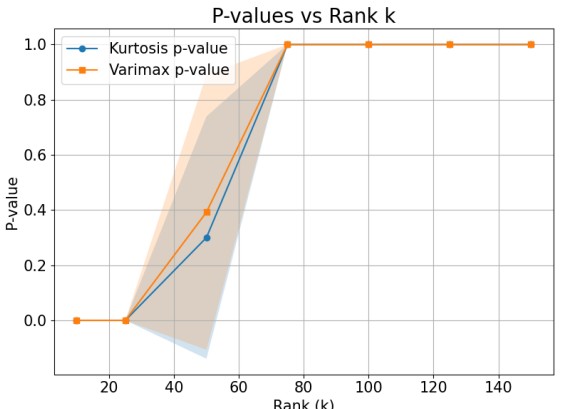 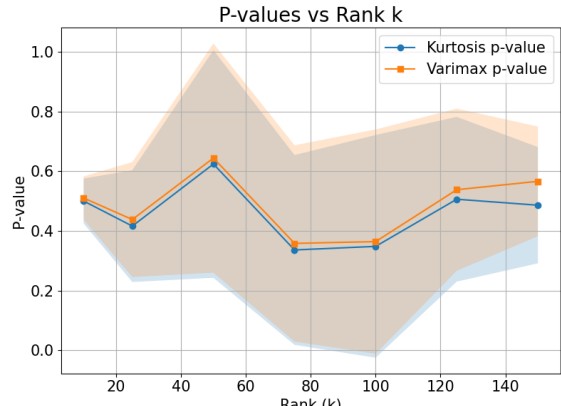

Figure 5: Illustration of how p-values change with rank $k$. Left: white noise image embedding from a pretrained ViT-L/14 model. Right: white noise embedding of dimension $10,000 \times 768$.

| Dataset | Task | Spurious Attribute | #Images | #Classes |
|---------|------|--------------------|---------|----------|
| Waterbirds | Bird Species | Background | 4,795 | 2 |
| iWildCam | Animal Species | Location | 42,791 | 182 |
| CelebA | Hair Color | Eyeglasses | 19,962 | 2 |

Table 4: Dataset characteristics and their corresponding spurious correlations.

## B.3 Spurious Concept Removal Experiment

**Datasets details**

- **Waterbirds** (Sagawa et al., 2019): Bird species (waterbird/landbird) with spurious background correlations (water/land)

- **WILDS-iWildCam** (Beery et al., 2020; Koh et al., 2021): Animal species classification with spurious location-specific features across camera traps

- **CelebA** (Liu et al., 2015): Hair color classification (blonde/non-blonde) with spurious attribute correlation (eyeglasses)

Dataset statistics can be found in table 4.

**Spurious Concept Detection Methods.** We develop strategies to automatically identify spurious concepts, tailored to each dataset's characteristics:

- For **Waterbirds**, we generate text descriptions following the template "A {bird_type} with a {background_type} background." We identify spurious concepts as those where top-ranking descriptions share common backgrounds but varied bird species.

- For **iWildCam**, we employ a contrastive approach using two sets of descriptions: one focusing on animal features and another on location attributes. We compute cosine similarities between concept embeddings and these description embeddings to identify concepts that correlate strongly with location features.

- For **CelebA**, we generate descriptions emphasizing either hair color or eyeglasses attributes, using a similar contrastive approach to separate target concepts from spurious ones.

**Removing Spurious Concepts** To remove spurious concepts, we reconstruct image embeddings while setting the coefficients of identified spurious concepts to zero. Given the decomposition $A_{i.} = \sum_j \alpha_j C_j$ where $C_j$ are learned concept vectors, we enforce $\alpha_{spurious} = 0$ to filter out spurious information.

**Waterbirds Experiment** For the Waterbirds experiment, we use ['a landbird', 'a waterbird'] as class prompts. In the zero-shot prediction experiment, we first compute text embeddings for class prompts and compute cosine similarity between class prompt embeddings and image embeddings. Then, for each image, we extract the class with the highest similarity as the prediction.

For removing spurious concepts, we first decompose image embedding into a linear combination of concepts with Algorithm 2: $A_{i.} = \sum_j \alpha_j C_j$. Suppose we have identified spurious concepts with our proposed method, as explained in the main content. By removing spurious concepts, we set the coefficients for spurious concepts to 0. In other words, $\alpha_{spurious} = 0$.

### B.4 Algorithm to Generate Rotation-Invariant Matrix

See algorithm to generate Rotation-Invariant matrix in Algorithm 4.

---

**Algorithm 4** Generate Rotation-Invariant Matrix

---

1: **Input:** Matrix $U \in \mathbb{R}^{n \times k}$
2: **Output:** Rotation-invariant matrix $U^{\text{rot}} \in \mathbb{R}^{n \times k}$
3: **for** each row $u_i \in \mathbb{R}^k$, $i = 1, 2, \ldots, n$ **do**
4:      Generate random rotation matrix $R_i \in \mathbb{R}^{k \times k}$
5:      $u_i^{\text{rot}} \leftarrow R_i u_i$                                                      ▷ Rotate the row $u_i$
6: **end for**
7: $U^{\text{rot}} \leftarrow [u_1^{\text{rot}}, u_2^{\text{rot}}, \ldots, u_n^{\text{rot}}]^T$                               ▷ Matrix of rotated rows
8: **return** $U^{\text{rot}}$

---

## C Additional Experiment Results

### C.1 Bootstrap simulation results

In Figure 6, we present bootstrap kurtosis and observed kurtosis distribution defined in Section 3.3.

### C.2 Additional Concept Results for ImageNet

We provide additional concept results for the ImageNet validation set in Figure 7. Embeddings are computed by the ViT-B-32 model.

### C.3 Concept Results for Waterbirds

We provide concept results for the Waterbirds dataset in Figure 8.

### C.4 Generalization Across Model Architectures

To demonstrate that our framework is not constrained to a specific architecture, we evaluate it on models with different backbones and training data: ResNet-50 (CNN-based, OpenAI weights) and OpenCLIP ViT-B-32 trained on LAION-2B. Across all tested models, we observe a consistent pattern: concepts derived from raw SVD (without rotation) frequently appear as entangled mixtures of unrelated themes. However, after applying our proposed Varimax rotation, the directions consistently align into semantically unified themes. When applied to the same dataset (ImageNet), different backbones (e.g., ResNet-50 vs. ViT) yield similar semantic concepts, identifying shared themes such as dogs, knitwear, birds, and ships. These results confirm that the rotation-sensitive structure is not an artifact of the ViT architecture but a general property of CLIP-style contrastive embeddings.

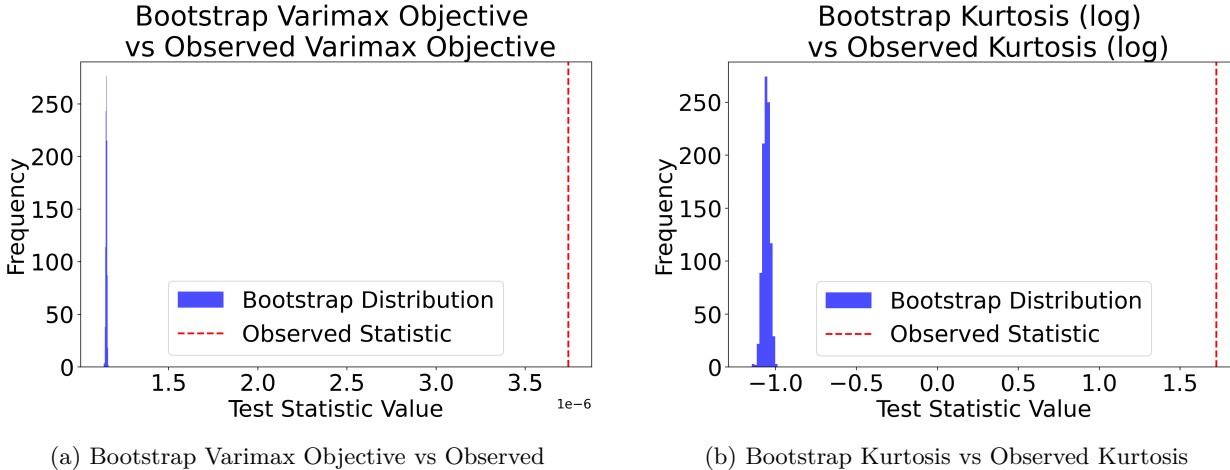

(a) Bootstrap Varimax Objective vs Observed     (b) Bootstrap Kurtosis vs Observed Kurtosis

Figure 6: Comparison of bootstrap distributions and observed test statistics. The blue histograms show the distribution of test statistics computed from rotation-invariant resamples under the null hypothesis. The red dashed lines indicate the observed test statistics computed from CLIP embeddings of the ImageNet validation set.

## C.5 Robustness under Random Orthogonal Rotations.

To further validate our framework, we evaluate how different methods behave when the CLIP image embedding space is randomly rotated. We sample a random orthogonal matrix $Q \sim \text{Uniform}(\text{SO}_d)$ and apply it to all image embeddings: $\tilde{A} = AQ$. We then compare three methods on the rotated embeddings:

- **Our method** maintains high interpretability and low reconstruction error on the rotated data, confirming that it successfully recovers the underlying structure even under arbitrary rotation.
- **SpLiCE**, which relies on projecting embeddings onto a fixed concept vocabulary, catastrophically fails: the rotation breaks the alignment between image and text embeddings, resulting in a large increase in reconstruction error and a loss of semantic relevance.
- **Raw SVD** is robust to the rotation transformation since singular vectors rotate with the data. However, the resulting concepts remain mixed and polysemantic, demonstrating that variance-based decomposition alone is insufficient—an explicit search for the "simple structure" rotation is required to disentangle semantic concepts.

Specifically, we sample random orthogonal matrices $Q \sim \text{Uniform}(\text{SO}_d)$ and apply them to CLIP ViT-B/32 image embeddings: $\tilde{A} = AQ$. We then run each method (Ours, SpLiCE, Raw SVD) on the rotated embeddings. For our method and Raw SVD, we compute the SVD of the rotated data and apply Varimax rotation (or not, for Raw SVD). For SpLiCE, we project the rotated embeddings onto the original fixed text concept vocabulary. We track two metrics: (1) reconstruction error (mean squared error between original and reconstructed embeddings) and (2) concept coherence score (average pairwise cosine similarity among top-9 activating images per concept). The experiment was repeated with 5 different random seeds to account for variability in the rotation sampling.

## C.6 Quantitative Concept Coherence Results

To quantify the "semantic purity" of discovered concepts, we introduce a *concept coherence score*. For each concept direction, we retrieve the top-9 most activating images and compute the average pairwise cosine similarity between their embeddings. A higher score indicates a tight, consistent semantic cluster, while a lower score indicates polysemantic concepts. As shown in Table 5, our method achieves the highest coherence across all tested models and training configurations, confirming it minimizes concept entanglement better than both ICA and Raw SVD.

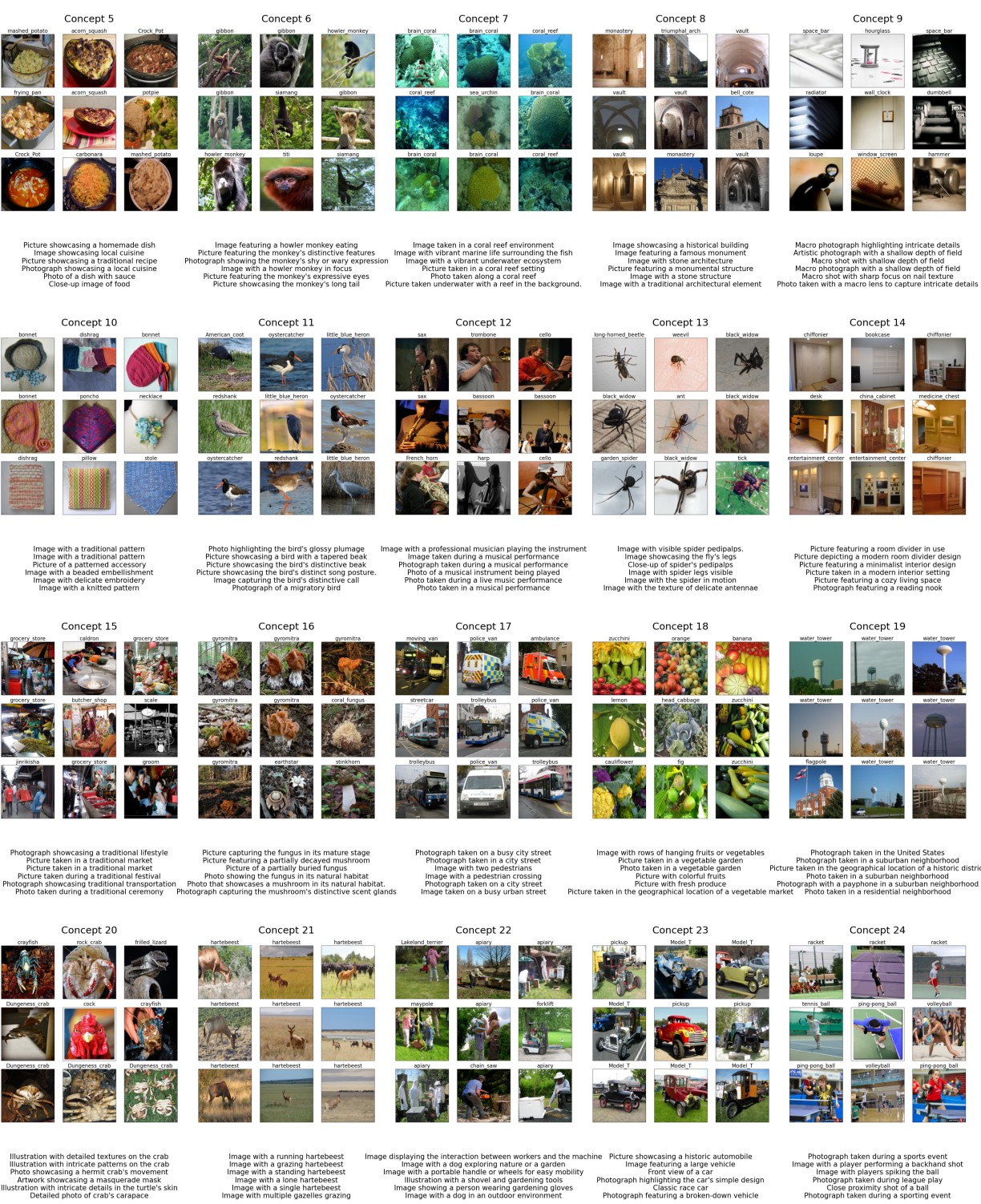

Figure 7: Top-24 concepts using our method with leading images and corresponding text descriptions. We observe that image and text concepts are well-aligned with similar semantic topics.

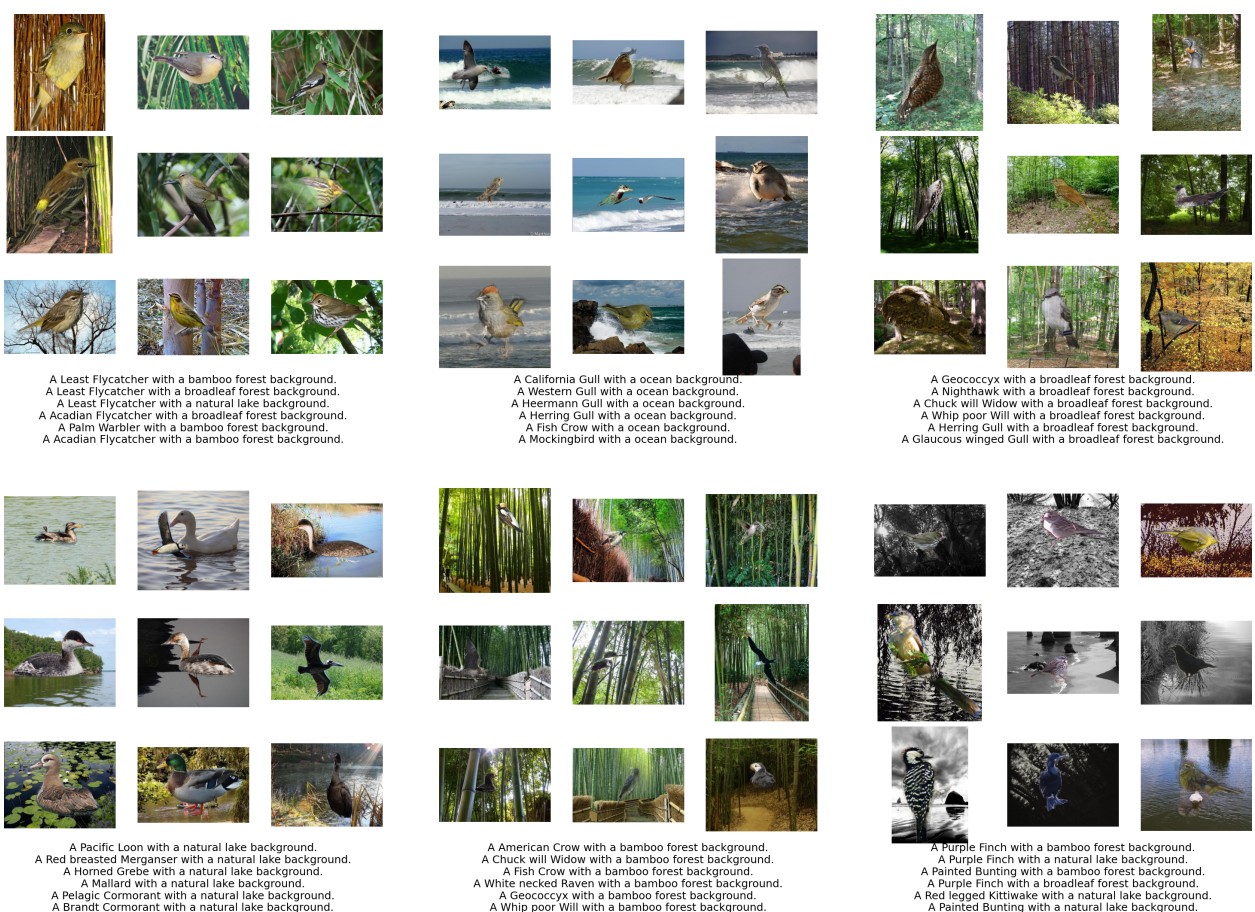

Figure 8: Top-6 Waterbrids dataset concepts with text descriptions. We noticed there are bird-focused concepts (e.g., first row, left column) that specify the species more clearly and mention distinctive features. There are background-focused concepts (e.g., first row, middle column), that highlight the type of environment. We also observed a multiple birds concept (second row, left column).

Table 5: Quantitative Concept Coherence Scores across different CLIP models. Higher scores indicate more semantically coherent concepts. Best results in **bold**.

| Model | Weights | ICA | Raw SVD | Ours |
|---|---|---|---|---|
| ViT-B-32 | openai | 0.649 | 0.621 | **0.764** |
| RN50 | openai | 0.613 | 0.604 | **0.744** |
| ViT-L-14 | openai | 0.643 | 0.601 | **0.725** |
| ViT-B-32 | laion2b_s34b_b79k | 0.578 | 0.499 | **0.677** |

Table 6: Spurious-Concept Removal Performance on Waterbirds. All concept-based methods use $k = 50$ concepts. "Recon" denotes reconstruction from all concepts; "Spurious Removed" denotes reconstruction after removing identified spurious concepts. Best unsupervised results in **bold**.

| Method | Recon Avg | Recon WG | Removed Avg | Removed WG |
|---|---|---|---|---|
| ERM (Supervised) | — | — | 94.9% | 29.2% |
| NMF | 25.3% | 0.0% | 23.0% | 0.1% |
| ICA | 86.5% | 40.8% | 80.6% | 52.1% |
| CRAFT | 23.9% | 0.0% | 22.6% | 0.0% |
| SVD | 85.5% | 39.0% | 70.7% | 60.4% |
| **Ours** | 85.5% | 39.0% | **89.6%** | **60.7%** |

## C.7 Additional Concept Removal Experiment Results for Waterbirds Dataset

To further contextualize our method, we compare against a broader set of baselines on the Waterbirds dataset, including both supervised and unsupervised approaches.

**Setup.** We evaluate the following methods, all using $k = 50$ concepts where applicable: a supervised linear probe (ERM) trained with ground-truth labels, non-negative matrix factorization (NMF), independent component analysis (ICA), Sparse Autoencoder (Sparse-AE) (Cunningham et al., 2023), and CRAFT (Fel et al., 2023). Performance is measured by zero-shot classification accuracy after reconstructing embeddings with and without spurious components removed.

**Results.** Table 6 confirms the intuition regarding the robustness-accuracy trade-off. The supervised ERM baseline achieves the highest average accuracy (94.9%) but is fragile, collapsing to 29.2% on the worst-group. NMF and CRAFT fail to produce meaningful reconstructions, yielding near-zero worst-group accuracy. ICA achieves moderate performance but our method outperforms it on both average accuracy (89.6% vs. 80.6%) and worst-group accuracy (60.7% vs. 52.1%) after spurious concept removal. Our method sacrifices some average performance compared to ERM to gain substantial robustness, verifying that we successfully identify and remove the spurious background concepts that the supervised baseline relies on.

## C.8 Connection to Internal Model Mechanisms.

To investigate whether our global concepts reflect internal model mechanisms, we analyze the alignment between our learned concept directions and the attention heads of the model's last layer. Following Gandelsman et al. (2024a), who showed that CLIP's representation is primarily constructed in the late attention layers, we computed the cosine similarity between our concept directions and the outputs of the last-layer attention heads. The resulting alignment is sparse and structured: specific global concepts align strongly with particular attention heads, indicating functional specialization. This suggests our concepts are not arbitrary directions but directly reflect internal model mechanisms.

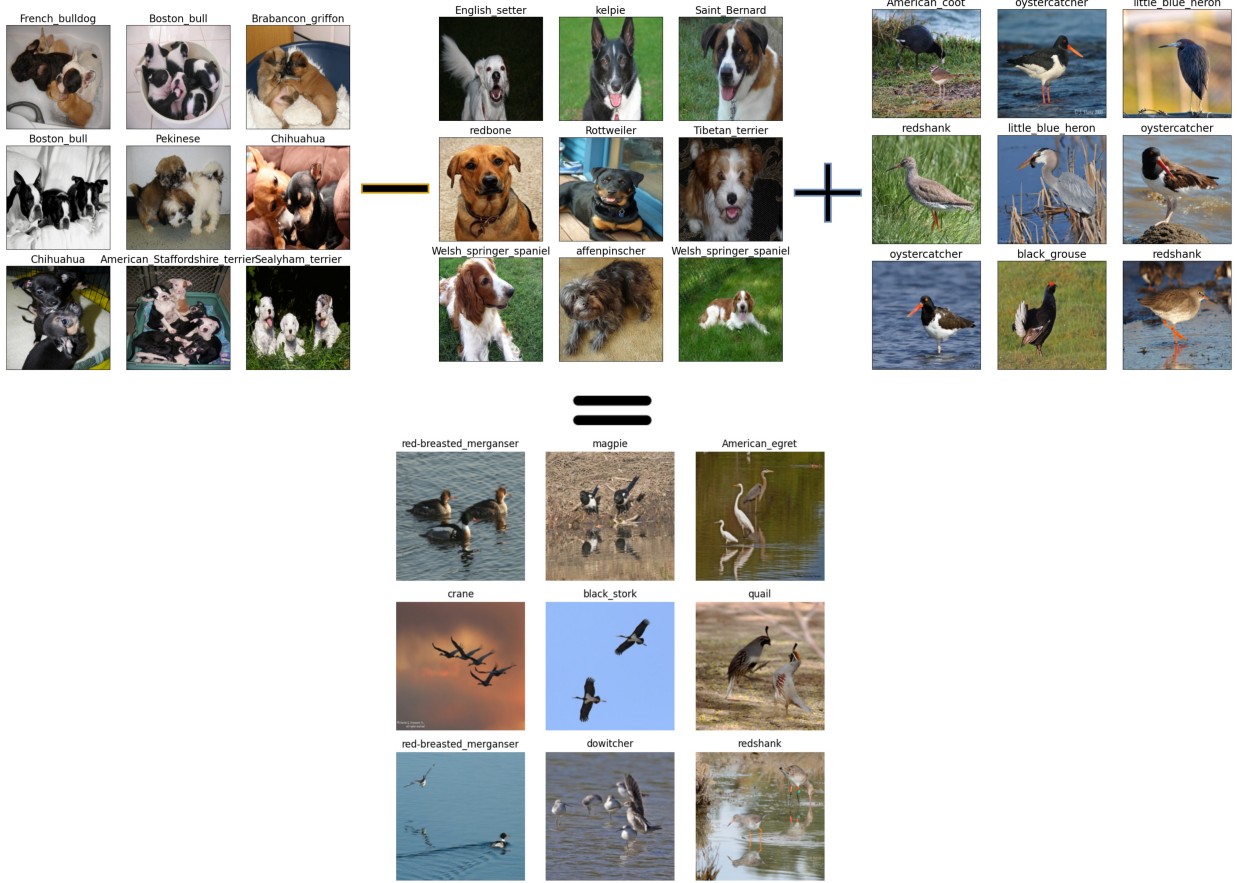

Figure 9: Demonstration of analogical reasoning with concepts. The equation $C_{gd}$ (group of dogs) $- C_d$ (single dog) $+ C_b$ (single bird) yields a concept that correctly identifies groups of birds in both image and text spaces.

### C.9   Concept Learns Analogical Relations

Word embeddings are known to capture semantic relationships through vector arithmetic, famously demonstrated by analogies such as "king - man + woman = queen" (Mikolov et al., 2013). We demonstrate that our learned concepts exhibit similar compositional properties with visual concepts.

To evaluate this, we identify three key concepts from our learned representation: $C_{gd}$ (representing groups of dogs), $C_d$ (single dog), and $C_b$ (bird). We then construct a new concept through vector arithmetic: $C = C_{gd} - C_d + C_b$. Intuitively, this operation should capture the transformation from "single entity" to "group" and apply it to birds. We evaluate this constructed concept in two ways: by projecting image embeddings ($Score = AC$ where $A$ contains image embeddings) and text embeddings onto this concept space. As shown in Figure 9, both the top-scoring images and their associated text descriptions align with our expectation, consistently returning groups of birds, demonstrating that our method successfully captures and transfers the concept of collectiveness across different semantic categories.

## D   Technical Lemmas

In this section, we provide some technical results for convenience.

The following lemma is a generalization of Li et al. (2023, Lemma H.5) for a non-square matrix. We recall $\sigma_k(Z) = \sqrt{\sigma_k(Z^\top Z)}$ as the $k$-th largest absolute singular value of $Z$.

**Lemma 1.** *For $\mathbf{A} \in \mathbb{R}^{d \times k}$, $\mathbf{B} \in \mathbb{R}^{k \times n}$ where $n > d \geq k$, we have*

$$\|\mathbf{A}\|_{\mathrm{F}} \cdot \sigma_k(\mathbf{B}) \leq \|\mathbf{AB}\|_{\mathrm{F}} \leq \|\mathbf{A}\|_{\mathrm{F}} \cdot \sigma_1(\mathbf{B}). \tag{2}$$

*Proof.* Assume the SVD of $\mathbf{B}$ is $U_{\mathbf{B}} D_{\mathbf{B}} V_{\mathbf{B}}$. Then,

$$\|\mathbf{AB}\|_{\mathrm{F}} = \|\mathbf{A} U_{\mathbf{B}} D_{\mathbf{B}} V_{\mathbf{B}}\|_{\mathrm{F}} = \|\mathbf{A} U_{\mathbf{B}} D_{\mathbf{B}}\|_{\mathrm{F}}. \tag{3}$$

By applying a similar induction proof as in Li et al. (2023, Lemma H.5) to $\mathbf{A} U_{\mathbf{B}} D_{\mathbf{B}}$ where $\mathbf{A} U_{\mathbf{B}} \in \mathbb{R}^{d \times k}$ and $D_{\mathbf{B}} \in \mathbb{R}^{k \times k}$, we obtain

$$\|\mathbf{A} U_{\mathbf{B}} D_{\mathbf{B}}\|_{\mathrm{F}} \geq \|\mathbf{A} U_{\mathbf{B}}\|_{\mathrm{F}} \cdot \sigma_k(D_{\mathbf{B}}) = \|\mathbf{A}\|_{\mathrm{F}} \cdot \sigma_k(D_{\mathbf{B}}) \tag{4}$$

and

$$\|\mathbf{A} U_{\mathbf{B}} D_{\mathbf{B}}\|_{\mathrm{F}} \leq \|\mathbf{A} U_{\mathbf{B}}\|_{\mathrm{F}} \cdot \sigma_1(D_{\mathbf{B}}) = \|\mathbf{A}\|_{\mathrm{F}} \cdot \sigma_1(D_{\mathbf{B}}). \tag{5}$$

This concludes the proof. $\square$

## E  Missing proofs

### E.1  Proof of Example 1

*Proof.* The density of the standard multivariate normal distribution is given by

$$f(x) = \frac{1}{(2\pi)^{d/2}} \exp\left(-\frac{1}{2}\|x\|^2\right),$$

which depends only on the $L_2$ norm $\|x\|$. For any rotation matrix $R$, we have $\|xR\| = \|x\|$ because rotations preserve norms. Therefore,

$$f(xR) = \frac{1}{(2\pi)^{d/2}} \exp\left(-\frac{1}{2}\|xR\|^2\right) = \frac{1}{(2\pi)^{d/2}} \exp\left(-\frac{1}{2}\|x\|^2\right) = f(x).$$

This shows that the density (and thus the distribution) of $x$ is unchanged under rotations, proving rotational invariance. $\square$

### E.2  Proof of Example 2

*Proof.* It suffices to show the result when $k = \min\{n, m\}$. Due to the nature of normally distributed random variables, for any orthogonal matrices $G \in \mathbb{R}^{m \times m}$ and $H \in \mathbb{R}^{n \times n}$, the entries of $GAH$ are i.i.d. and follow a standard normal distribution. Therefore, this guarantees spherical symmetry for the left and right singular matrices $U$ and $V^\top$ of $A$, implying that both $U$ and $V^\top$ follow a uniform distribution with respect to the Haar measure on the Stiefel manifold.

For any rotation matrix $R \in \mathbb{R}^{k \times k}$, we have $(UR)^\top UR = \mathcal{I}_k$. Consider $f_R(U) = UR$, which defines a one-to-one map from the Stiefel manifold to itself. By the one-to-one property, $UR$ has the same distribution as $U$, namely following the uniform distribution to the Haar measure on the Stiefel manifold. A similar result holds for $V^\top$.

This guarantees rotation invariance. $\square$

### E.3  Proof of Example 3

*Proof.* To show that this distribution is not rotationally invariant, we need to find a rotation matrix $R$ such that the distribution of $xR$ differs from the distribution of $x$.

Let $R$ be a rotation matrix that rotates the vector $\mu$ to another direction. For simplicity, consider a rotation $R$ that maps $\mu$ to $R\mu = \nu$, where $\nu = (0, 1, 0, \ldots, 0)^\top$.

The original distribution of $x$ has two components centered at $\mu$ and $-\mu$. After rotation, the distribution of $xR$ has components centered at $R\mu = \nu$ and $R(-\mu) = -\nu$.

However, the probability density function (pdf) of $x$ before rotation is

$$f(x) = \frac{1}{2}\frac{1}{(2\pi)^{d/2}}\exp\left(-\frac{1}{2}\|x-\mu\|^2\right) + \frac{1}{2}\frac{1}{(2\pi)^{d/2}}\exp\left(-\frac{1}{2}\|x+\mu\|^2\right).$$

After rotation, the PDF becomes

$$f_R(x) = f(xR) = \frac{1}{2}\frac{1}{(2\pi)^{d/2}}\exp\left(-\frac{1}{2}\|xR-\mu\|^2\right) + \frac{1}{2}\frac{1}{(2\pi)^{d/2}}\exp\left(-\frac{1}{2}\|xR+\mu\|^2\right).$$

But since $\|xR-\mu\|^2 \neq \|x-\mu\|^2$ in general, the pdf $f_R(x)$ is not equal to $f(x)$. Specifically, the locations of the mixture components have changed, resulting in a different distribution.

Moreover, consider evaluating the probability at a specific point. For example, at $x = \mu$, we have

$$\begin{aligned} f(\mu) &= \frac{1}{2(2\pi)^{d/2}}\exp\left(-\tfrac{1}{2}\|\mu-\mu\|^2\right) + \frac{1}{2(2\pi)^{d/2}}\exp\left(-\tfrac{1}{2}\|\mu+\mu\|^2\right) \\ &= \frac{1}{2(2\pi)^{d/2}}\left[1 + \exp\left(-2\|\mu\|^2\right)\right]. \end{aligned}$$

After rotation, at $x = \mu$, we have

$$f_R(\mu) = f(\mu R) = \frac{1}{2}\frac{1}{(2\pi)^{d/2}}\exp\left(-\frac{1}{2}\|\mu R-\mu\|^2\right) + \frac{1}{2}\frac{1}{(2\pi)^{d/2}}\exp\left(-\frac{1}{2}\|\mu R+\mu\|^2\right).$$

Since $\mu R \neq \mu$, the values of $f(\mu)$ and $f_R(\mu)$ are different, confirming that the distribution is not rotationally invariant. □

### E.4 Proof of Proposition 1

Under $H_0$, to show that $A^{\mathrm{rot}}$ is rotation invariant, we need to prove that for any fixed rotation matrix $R \in \mathrm{SO}(d)$, the distribution of $A^{\mathrm{rot}}R$ is the same as that of $A^{\mathrm{rot}}$.

Since conditional on the same $\|A_i\|_2 = \|A_i^{\mathrm{rot}}\|_2$, for any $A_i^{\mathrm{rot}}$ there must exist $R_i$ such that

$$A_i^{\mathrm{rot}} = A_i R_i,$$

where $R_i$ uniformly sampled from $\mathrm{SO}(d)$. For any $R \in \mathrm{SO}(d)$, multiplying $A^{\mathrm{rot}}$ on the right by $R$:

$$A_i^{\mathrm{rot}}R = (A_i R_i)R = A_i(R_i R) = A_i \tilde{R}_i,$$

where we define $\tilde{R}_i = R_i R$. Since $R_i$ are uniformly distributed over $\mathrm{SO}(d)$ and independent, and $R$ is a fixed element of $\mathrm{SO}(d)$, the products $\tilde{R}_i = R_i R$ are also uniformly distributed over $\mathrm{SO}(d)$, independent from each other, and independent from $A_i$. Therefore, the distribution of $A_i \tilde{R}_i$ is the same as that of $A_i R_i$:

$$A_i \tilde{R}_i \overset{d}{=} A_i R_i.$$

This implies that the rows of $A^{\mathrm{rot}}R$ have the same joint distribution as the rows of $A^{\mathrm{rot}}$:

$$\{A_i^{\mathrm{rot}}R\}_{i=1}^n \overset{d}{=} \{A_i^{\mathrm{rot}}\}_{i=1}^n.$$

This completes the proof.

### E.5 Proof of Theorem 1

For theoretical analysis, we derive an equivalent standardized form:

$$TS_3(U) = \frac{\sqrt{nk}}{\sqrt{33}} \left( \frac{1}{k} \sum_{i=1}^{k} |\text{kurtosis}(U_{\cdot i})| - \frac{3n}{n+2} \right). \tag{6}$$

Recall that $U^\top U = \mathbf{I}$, therefore we have $\sum_{j=1}^{n} U_{ji}^2 = 1$ for any $i$. We can simplify the rescaled kurtosis as

$$\text{TS}_3(U_{\cdot i}) = \frac{\sqrt{nk}}{\sqrt{33}} \left( \frac{n}{k} \sum_{i=1}^{k} \sum_{j=1}^{n} U_{ji}^4 - \frac{3n}{n+2} \right). \tag{7}$$

We recall from the proof of Example 2 that for fixed $i$, $U_{\cdot i}$ follows a normal distribution on Haar measure. Denote

$$X_i = n \sum_{j=1}^{n} U_{ji}^4 - \frac{3n}{n+2}.$$

We compute

$$\mathbb{E}[U_{ij}^{2s}] = \frac{\Gamma(\frac{n}{2})\Gamma(s+\frac{1}{2})}{\Gamma(\frac{n}{2}+s)\Gamma(\frac{1}{2})}.$$

When $s = 2$, we have

$$\mathbb{E}[U_{ij}^4] = \frac{3}{n(n+2)}.$$

Therefore, by plugging this formula into our computation, we obtain

$$\mathbb{E}[X_i] = \frac{3n}{n+2} - \frac{3n}{n+2} = 0,$$

and

$$\mathbb{E}[U_{ij}^8] = \frac{\frac{7}{2}\frac{5}{2}\frac{3}{2}\frac{1}{2}}{(\frac{n}{2}+3)(\frac{n}{2}+2)(\frac{n}{2}+1)\frac{n}{2}}.$$

On the other hand, by rewriting $U_{ij}$ as $\frac{S_i}{\sqrt{\sum_i S_i^2}}$, we obtain $(U_{ij}, U_{i'j})$ and $(\frac{U_{ij}+U_{i'j}}{\sqrt{2}}, \frac{U_{ij}-U_{i'j}}{\sqrt{2}})$ are identically distributed. Therefore, we have

$$\mathbb{E}\left[ U_{ij}^4 U_{i'j}^4 \right] = \mathbb{E}\left[ \left( \frac{U_{ij}+U_{i'j}}{\sqrt{2}} \right)^4 \left( \frac{U_{ij}-U_{i'j}}{\sqrt{2}} \right)^4 \right].$$

This can be reduced to

$$\mathbb{E}\left[ U_{ij}^8 \right] = 4\mathbb{E}\left[ U_{ij}^6 U_{i'j}^2 \right] + 5\mathbb{E}\left[ U_{ij}^4 U_{i'j}^4 \right]. \tag{8}$$

On the other hand, we have

$$\mathbb{E}\left[ U_{i'j}^6 U_{i'j}^2 \right] = \frac{1}{n-1} \left( \mathbb{E}\left[ U_{i'j}^6 \right] - \mathbb{E}\left[ U_{i'j}^6 \right] \right) = \frac{15}{(n+6)(n+4)(n+2)n}. \tag{9}$$

Combining equation 8 and equation 9, we obtain,

$$\mathbb{E}\left[ U_{ij}^4 U_{i'j}^4 \right] = \frac{9}{(n+6)(n+4)(n+2)n}.$$

By plugging in the computations of moments, we obtain

$$\mathbb{E}\left[ X_i^2 \right] = n^3 \mathbb{E}[U_{ij}^8] + n^3(n-1)\mathbb{E}[U_{ij}^4 U_{i'j}^4] - n^3(n-1)(\mathbb{E}[U_{ij}^4])^2$$

$$= \frac{105 \times n^2}{(n+6)(n+4)(n+2)} + \frac{9n^2(n-1)}{(n+6)(n+4)(n+2)} - \frac{9n(n-1)}{(n+2)^2}.$$

The leading order term of this variance is $\frac{33}{n}$. By the central limit theorem and the fact that $X_i^2$ are i.i.d. random variables, we conclude the result.

### E.6 Proof of Theorem 2

Recall Assumption 1 that $\mathbb{E}[\tilde{Z}_{ij}] = 0$, $\mathbb{E}(\tilde{Z}_{ij}^2) = \sigma_j^2$, $\mathbb{E}(\tilde{Z}_{ij}^4) = \eta_j \geq 3\sigma_j^4$.

$$v(R, \tilde{Z}\tilde{R}^\top) = \frac{1}{n}\sum_{\ell=1}^k \sum_{i=1}^n \left([\tilde{Z}\tilde{R}R]_{i\ell}^4 - \left(\frac{1}{n}\sum_{q=1}^n [\tilde{Z}\tilde{R}R]_{q\ell}^2\right)^2\right).$$

To simplify notation, we denote $O = \tilde{R}R \in \mathcal{O}(k)$. We want to optimize $v(R, \tilde{Z}^\top \tilde{R}^\top)$ over $O$. We analyze two terms, respectively. For the fourth moment term

$$
\begin{aligned}
\mathbb{E}\left(\frac{1}{n}\sum_{\ell=1}^k \sum_{i=1}^n [\tilde{Z}O]_{i\ell}^4\right) &= \mathbb{E}\left(\frac{1}{n}\sum_{\ell=1}^k \sum_{i=1}^n \left(\sum_{j=1}^k \tilde{Z}_{ij}O_{jl}\right)^4\right) \\
&= \mathbb{E}\left(\frac{1}{n}\sum_{\ell=1}^k \sum_{i=1}^n \left(\frac{1}{n}\sum_{\ell=1}^k \sum_{i=1}^n \tilde{Z}_{ij}^4 O_{jl}^4 + 3\sum_{h\neq h'} \tilde{Z}_{ih}^2 \tilde{Z}_{ih'}^2 O_{hl}^2 O_{h'l}^2\right)\right) \\
&= \frac{1}{n}\sum_{\ell=1}^k \sum_{i=1}^n \left(\sum_{j=1}^k \eta_j O_{jl}^4 + 3\sum_{h\neq h'} \sigma_h^2 \sigma_h'^2 O_{hl}^2 O_{h'l}^2\right) \\
&= \sum_{\ell=1}^k \left(\sum_{j=1}^k \eta_j O_{jl}^4 + 3\sum_{h\neq h'} \sigma_h^2 \sigma_h'^2 O_{hl}^2 O_{h'l}^2\right),
\end{aligned}
$$

because the expectation of all other cross terms in the computation contains at least one moment of an entry, which is 0 according to the independence and $\mathbb{E}[\tilde{Z}_{ij}] = 0$. For the second moment term,

$$
\begin{aligned}
\mathbb{E}\left[\frac{1}{n}\sum_{\ell=1}^k \sum_{i=1}^n \left(\frac{1}{n}\sum_{q=1}^n [\tilde{Z}O]_{q\ell}^2\right)^2\right] &= \frac{1}{n^2}\sum_{\ell=1}^k \left(\sum_{q=1}^n \left(\sum_{j=1}^k \tilde{Z}_{qj}O_{jl}\right)^2\right)^2 \\
&= \mathbb{E}\left[\frac{1}{n^2}\sum_{\ell=1}^k \left(\sum_{q=1}^n \sum_{j=1}^k \tilde{Z}_{qj}^2 O_{jl}^2 + \sum_{q=1}^n \sum_{h\neq h'} \tilde{Z}_{qh}\tilde{Z}_{qh'}O_{hl}O_{h'l}\right)^2\right] \\
&= \frac{1}{n^2}\text{ cross terms} + \underbrace{\mathbb{E}\left[\frac{1}{n^2}\sum_{\ell=1}^k \left(\sum_{q=1}^n \sum_{j=1}^k \tilde{Z}_{qj}^2 O_{jl}^2\right)^2\right]}_{\text{Term 1}} \\
&\qquad + \underbrace{\mathbb{E}\left[\frac{1}{n^2}\sum_{\ell=1}^k \left(\sum_{q=1}^n \sum_{h\neq h'} \tilde{Z}_{qh}\tilde{Z}_{qh'}O_{hl}O_{h'l}\right)^2\right]}_{\text{Term 2}}.
\end{aligned}
$$

Since $\mathbb{E}(\tilde{Z}_{qj}) = 0$, the expectation of all the cross terms should be 0 and can be removed from the equation. Now we compute terms 1 and 2, respectively. For term 1,

$$
\begin{aligned}
\mathbb{E}\left(\text{Term 1}\right) &= \frac{1}{n^2} \sum_{\ell=1}^{k} \mathbb{E}\left( \sum_{q=1}^{n} \sum_{j=1}^{k} \tilde{Z}_{qj}^2 O_{jl}^2 \right)^2 \\
&= \frac{1}{n^2} \sum_{\ell=1}^{k} \left( \text{Var}\left( \sum_{q=1}^{n} \sum_{j=1}^{k} \tilde{Z}_{qj}^2 O_{jl}^2 \right) + \left( \mathbb{E}\left[ \sum_{q=1}^{n} \sum_{j=1}^{k} \tilde{Z}_{qj}^2 O_{jl}^2 \right] \right)^2 \right) \\
&= \frac{1}{n^2} \sum_{\ell=1}^{k} \left( \sum_{q=1}^{n} \sum_{j=1}^{k} O_{jl}^4 \text{Var}\left( \tilde{Z}_{qj}^2 \right) + \left( \sum_{q=1}^{n} \sum_{j=1}^{k} O_{jl}^2 \mathbb{E}(\tilde{Z}_{qj}^2) \right)^2 \right) \\
&= \sum_{\ell=1}^{k} \left( \frac{1}{n} \sum_{j=1}^{k} O_{jl}^4 (\eta_j - \sigma_j^4) + \left( \sum_{j=1}^{k} O_{jl}^2 \sigma_j^2 \right)^2 \right) \\
&= \sum_{\ell=1}^{k} \left( \sum_{j=1}^{k} O_{jl}^2 \sigma_j^2 \right)^2 + \frac{1}{n} \sum_{\ell=1}^{k} \sum_{j=1}^{k} (\eta_j - \sigma_j^4) O_{jl}^4.
\end{aligned}
$$

For term 2,

$$
\begin{aligned}
\mathbb{E}\left(\text{Term 2}\right) &= \frac{1}{n^2} \sum_{\ell=1}^{k} \mathbb{E}\left( \sum_{q=1}^{n} \sum_{h \neq h'} \tilde{Z}_{qh} \tilde{Z}_{qh'} O_{hl} O_{h'l} \right)^2 \\
&= \frac{2}{n^2} \sum_{\ell=1}^{k} \sum_{q=1}^{n} \sum_{h \neq h'} \left( \mathbb{E}(\tilde{Z}_{qh}^2) \mathbb{E}(\tilde{Z}_{qh'}^2) O_{hl}^2 O_{h'l}^2 \right) \\
&= \frac{2}{n} \sum_{\ell=1}^{k} \sum_{h \neq h'} \left( \sigma_h^2 \sigma_{h'}^2 O_{hl}^2 O_{h'l}^2 \right) \\
&= \frac{2}{n} \sum_{\ell=1}^{k} \left( \left( \sum_{h=1}^{k} \sigma_h^2 O_{hl}^2 \right)^2 - \sum_{h=1}^{k} \sigma_h^4 O_{hl}^4 \right).
\end{aligned}
$$

Combining the computation for the second and fourth moments, we obtain

$$v\big(R, \tilde{Z}\,\tilde{R}^\top\big) = \sum_{\ell=1}^{k}\Big(\sum_{j=1}^{k}\eta_j\,O_{jl}^4 + 3\sum_{h\neq h'}\sigma_h^2\,\sigma_{h'}^2\,O_{hl}^2\,O_{h'l}^2\Big)$$

$$-\sum_{\ell=1}^{k}\Big(\sum_{j=1}^{k}O_{jl}^2\,\sigma_j^2\Big)^2 - \frac{1}{n}\sum_{\ell=1}^{k}\sum_{j=1}^{k}(\eta_j - \sigma_j^4)\,O_{jl}^4$$

$$-\frac{2}{n}\sum_{\ell=1}^{k}\Big(\big(\sum_{h=1}^{k}\sigma_h^2\,O_{hl}^2\big)^2 - \sum_{h=1}^{k}\sigma_h^4\,O_{hl}^4\Big)$$

$$= \sum_{\ell=1}^{k}\sum_{j=1}^{k}\Big(\tfrac{n-1}{n}\,\eta_j - \tfrac{3n-3}{n}\,\sigma_j^4\Big)O_{jl}^4 + \Big(2 - \tfrac{2}{n}\Big)\sum_{\ell=1}^{k}\Big(\sum_j \sigma_j^2\,O_{jl}^2\Big)^2$$

$$\leq \sum_{\ell=1}^{k}\sum_{j=1}^{k}\Big(\tfrac{n-1}{n}\,\eta_j - \tfrac{3n-3}{n}\,\sigma_j^4\Big) + \Big(2 - \tfrac{2}{n}\Big)\sum_{\ell=1}^{k}\sum_j \sigma_j^4 \sum_j O_{jl}^4$$

$$\leq \sum_{\ell=1}^{k}\sum_{j=1}^{k}\Big(\tfrac{n-1}{n}\,\eta_j - \tfrac{3n-3}{n}\,\sigma_j^4\Big) + \Big(2 - \tfrac{2}{n}\Big)\sum_{\ell=1}^{k}\sum_j \sigma_j^4.$$

Equality can and can only be achieved when $O$ is a permutation matrix, where each row and each column have and only have exactly one 1. This completes the proof.

## E.7 Proof of Theorem 3

Denote the generalized inverse of $\boldsymbol{C}_{\mathrm{W}}^\top \boldsymbol{C}_{\mathrm{W}}$ as $(\boldsymbol{C}_{\mathrm{W}}^\top \boldsymbol{C}_{\mathrm{W}})^\dagger$. Then, by projecting from the column space of $\boldsymbol{C}_{\mathrm{W}}$ to $\boldsymbol{C}^*$, we can rewrite the condition $\min_{P \in \mathbb{R}^{m \times k}} \|\boldsymbol{C}_{\mathrm{W}} P - \boldsymbol{C}^*\|_{\mathrm{F}} \geq \delta$ as

$$\|\boldsymbol{C}_{\mathrm{W}}(\boldsymbol{C}_{\mathrm{W}}^\top \boldsymbol{C}_{\mathrm{W}})^\dagger \boldsymbol{C}_{\mathrm{W}}^\top \boldsymbol{C}^* - \boldsymbol{C}^*\|_{\mathrm{F}} \geq \delta. \tag{10}$$

Similarly, by plugging $A = Z^* \boldsymbol{C}^{*\top}$,

$$\min_{Z \in \mathbb{R}^{n \times k}} \|A - Z \boldsymbol{C}_{\mathrm{W}}^\top\|_{\mathrm{F}} = \min_{Z \in \mathbb{R}^{n \times k}} \|Z^* \boldsymbol{C}^{*\top} - Z \boldsymbol{C}_{\mathrm{W}}^\top\|_{\mathrm{F}}$$

$$= \|Z^* \boldsymbol{C}^{*\top} - Z^* \boldsymbol{C}^{*\top} \boldsymbol{C}_{\mathrm{W}}(\boldsymbol{C}_{\mathrm{W}}^\top \boldsymbol{C})^\dagger \boldsymbol{C}_{\mathrm{W}}^\top\|_{\mathrm{F}}$$

$$\geq \|\boldsymbol{C}_{\mathrm{W}}(\boldsymbol{C}_{\mathrm{W}}^\top \boldsymbol{C}_{\mathrm{W}})^\dagger \boldsymbol{C}_{\mathrm{W}}^\top \boldsymbol{C}^* - \boldsymbol{C}^*\|_{\mathrm{F}} \cdot \sigma_k(Z^*),$$

where the inequality follows from Lemma 1. We conclude the result by plugging equation 10 in the above equation.

