# OpenReview forum: "Quantifying Structure in CLIP Embeddings: A Statistical Framework for Concept Interpretation"
_TMLR — Accepted by TMLR_

### Review · Reviewer_gnQX · 2025-12-23

**Summary Of Contributions:**

This paper proposes a concept-based framework for analyzing the CLIP embedding space by identifying directions that are sensitive to rotation and interpreting them as human-understandable concepts.
The proposed approach achieves a favorable trade-off between interpretability and reconstruction fidelity compared to prior methods that either prioritize interpretability through predefined semantic concepts (e.g., word-based decompositions) at the cost of reconstruction fidelity, or preserve embedding fidelity through SVD-based techniques while sacrificing interpretability.

The main contribution of the paper is the introduction of a statistical hypothesis testing framework to detect rotation-sensitive structure in embedding subspaces.
The key insight is that meaningful structure (i.e., concepts) should break rotational invariance, whereas noise remains statistically invariant under rotation.
Building on this idea, the paper applies Varimax rotation to align principal components with sparse, axis-aligned directions that are argued to be more interpretable, while maintaining low reconstruction error.
The authors further demonstrate that removing certain learned directions, identified as spurious, can improve accuracy on standard robustness benchmarks such as Waterbirds and iWildCam.

While the idea of rotation-sensitive structure is appealing, the paper does not carry the analysis far enough to meaningfully interpret the CLIP embedding space or the internal behavior of the model in the way earlier work does (e.g., Gandelsman et al., 2024a).
The proposed “concepts” are global and linear, and are not tied to model components, training dynamics, or representational mechanisms such as attention.
In contrast to earlier work that directly analyzes how CLIP representations are formed and used, the interpretation step here remains largely illustrative.
As a result, it is unclear how the extracted concepts advance understanding of CLIP’s internal dynamics, embedding geometry, or interactions between concepts beyond providing sparse decompositions with favorable reconstruction properties.

**Additional Comments:**

Minor comments:
- The language at some places sound informal such as  P.1 "CLIP is a powerful tool".
- The introduction of dimensionality $k$ in Example 2 in P.4 is abrupt without clarifying it to be the $k$ for truncated SVD.

**Audience:**

Yes

**Audience Explanation:**

While the paper introduces an interesting statistical perspective on rotation-sensitive structure in embedding spaces, it remains unclear how the proposed analysis meaningfully advances understanding of the CLIP embedding space itself.
The experimental evaluations rely on largely dataset-specific and ad-hoc setups, making it difficult to assess how the findings generalize or provide broader insight into CLIP representations.
Although the rotation-sensitivity analysis is potentially useful, the paper does not follow through with a deeper investigation of how the identified concepts arise within the model or how they relate to CLIP’s internal representational mechanisms.

**Broader Impact Concerns:**

There are not ethical implications of that work that needs to be addressed.

**Claims And Evidence:**

Yes

**Claims Explanation:**

In general, the paper provides sufficient evidence to support its core contribution on rotation-sensitive structure, including theoretical motivation and empirical validation on both synthetic noise and real CLIP embeddings.
The results convincingly show that CLIP embeddings from natural images are significantly more rotation-sensitive than white noise.
The reconstruction experiments and comparisons to existing methods are clearly presented and sufficient to substantiate claims about the interpretability/fidelity trade-off.

However, broader claims regarding interpretability are primarily supported by qualitative and illustrative examples.
The interpretation of rotation-sensitive directions as meaningful concepts relies on post-hoc alignment with images and text descriptions, without quantitative validation of semantic coherence or stability across datasets and models.
Similarly, while spurious correlation removal improves worst-group accuracy on benchmark datasets, it remains unclear whether these gains reflect deeper interpretability or simply the removal of dataset-specific nuisance features (e.g., background-related features in the Waterbirds dataset that are already known to be spurious).

Overall, the evidence is sufficient for the statistical methodology, but more limited for claims about the interpretability of the learned CLIP representations or concepts.

**Requested Changes:**

1- While the paper claims improved interpretability of the CLIP embedding space, it remains unclear how the proposed analysis leads to a deeper understanding of CLIP representations. Providing insights into how the identified rotation-sensitive directions relate to CLIP’s internal dynamics would help close this gap. Although not required for the current scope, connecting learned concept directions to architectural components, training signals, or representational mechanisms (e.g., attention patterns or layers) would significantly strengthen the interpretability narrative.

2- While the statistical detection of rotation-sensitive structure is convincing, the paper should more clearly explain why such structure should correspond to human-meaningful semantic concepts as claimed in the introduction section.

3- The spurious correlation experiments would benefit from a clearer discussion of whether the observed performance gains arise from meaningful disentanglement of semantic concepts or simply from removing known dataset-specific nuisance features (e.g. the background in Waterbirds dataset).

4- It would be great to get some insights on whether the proposed rotation-sensitivity analysis is expected to generalize to other vision language models. Clarifying the extent to which the approach is CLIP-specific versus model-agnostic would improve the positioning and relevance of the work.

---

> ### Author Response · Authors · 2026-01-25
> **Response to Reviewer gnQX (1/2)**
>
> We thank the reviewer for their thoughtful review, and for recognizing the soundness of our method and empirical results. We address all comments in detail below.
>
> * **On the connection to internal dynamics (Response to Request 1)**: Our framework is model-agnostic, focusing on final embedding geometry, but we agree that connecting these directions to internal components is valuable.
>     * **New Experiment:** To address this, we analyzed the alignment between our learned concepts and the attention heads of the model's last layer. We focused on the final layer following Gandelsman et al. (2024), who showed that CLIP's representation is primarily constructed in the late attention layers. We computed the cosine similarity between our concept directions and the outputs of the last-layer attention heads, and visualized it with a heatmap.
>     * **Results & Connection:** The heatmap shows a sparse, structured pattern: specific global concepts align strongly with particular heads, indicating functional specialization. This suggests our concepts are not arbitrary but directly reflect internal model mechanisms, and we will add this analysis in the revision.
>
> * **On the connection between rotation-sensitive structure and concept interpretability (Response to Request 2):** We agree that rotation sensitivity signals structure but does not by itself ensure interpretability. Our framework bridges this in two steps:
>     * **Detecting Structure:** First, our hypothesis testing framework uses rotation sensitivity to distinguish signal from noise. As shown in our random noise example, isotropic distributions are rotation-invariant (information is uniformly distributed). In contrast, rotation sensitivity reveals "preferred directions" in the embedding space, confirming the existence of a non-random, underlying data structure.
>     * **Aligning for Interpretability:** Once structure is detected, we use Varimax rotation to transform these directions into sparse concept loadings. This sparsity is the key to human interpretability: it associates data points only with their most relevant concepts and makes concepts distinct. Our results (Section 6.1) confirm that these sparse, rotated concepts exhibit clearer semantics than raw SVD components. This connection between sparsity and interpretability is well-established in representation learning literature [1-3], supporting our use of sparse rotations to extract meaningful definitions from the detected structure.
>
> * **On spurious correlations and meaningful disentanglement (Response to Request 3):** We argue that meaningful disentanglement and removing nuisance features are not mutually exclusive; rather, the ability to identify and remove a nuisance feature **is evidence of meaningful disentanglement.** In our Waterbirds experiment, the decomposition was unsupervised: when learning the concepts, the method did not know which axis was "background" and which was "bird." The fact that it isolated the background information into distinct, removable concepts (without destroying the bird classification information) demonstrates that the method successfully disentangled these two semantic factors into orthogonal directions. If the representation were not disentangled, removing the background axis would have degraded the core task performance.

---

> ### Author Response · Authors · 2026-01-25
> **Response to Reviewer gnQX (2/2)**
>
> * **On generalization to other Vision-Language Models (Response to Request 4):**
>     * It is correct that our framework is entirely model-agnostic and not strictly constrained to CLIP as it operates on the embedding space instead of model architecture. However, we initially focused on CLIP for two reasons:
>         * **Linear decodability:** CLIP is trained via a contrastive loss that maximizes the dot product between image and text embeddings. This objective explicitly enforces a linear structure: if a semantic concept exists, it must be linearly decodable in the shared embedding space. This makes CLIP ideal for verifying our rotation-based decomposition.
>         * **Automatic interpretation:** The aligned text encoder allows us to automatically interpret the discovered directions by projecting them onto text embeddings, rather than solely relying on human inspection of image cluster.
>     * To address the request for broader demonstration, we expand our evaluation to include models with different architectures and trained with different models, specifically we tested ResNet-50 (CNN-based) and OpenCLIP ViT-B-32 trained on LAION-2B.
>         * Across all tested models, we observed a consistent pattern: concepts derived from raw SVD (without rotation) frequently appeared as entangled mixtures. However, after applying our proposed rotation, the directions consistently aligned into semantically unified themes.
>         * When applied to the same dataset (ImageNet), different backbones (e.g., ResNet-50 vs. ViT) yielded similar semantic concepts to identify shared themes like dogs, knitwear, birds, and ships.
>         * These extracted concepts confirm that the rotation-sensitive structure is not an artifact of the ViT architecture. We have added these results to the appendix to demonstrate the method's broader applicability.
>
>
> **References**
>
> [1] Subramanian, Anant, et al. "Spine: Sparse interpretable neural embeddings." Proceedings of the AAAI conference on artificial intelligence. Vol. 32. No. 1. 2018.
>
> [2] Guillot, Simon, Thibault Prouteau, and Nicolas Dugué. "Sparser is better: one step closer to word embedding interpretability." Proceedings of the 15th International Conference on Computational Semantics. 2023.
>
> [3] Faruqui, Manaal et al. “Sparse Overcomplete Word Vector Representations.” Annual Meeting of the Association for Computational Linguistics (2015).

---

### Review · Reviewer_MqaU · 2025-12-23

**Summary Of Contributions:**

1. This paper clearly points out an issue underlying the trade‑off between interpretability and reconstruction fidelity — that existing methods lack a rigorous statistical framework and are inherently noisy.
2. Establishing a connection between rotational invariance and the interpretability and correctness of concepts is an interesting attempt.
3. The method for detecting rotational invariance is clear and has solid theoretical guarantees (although such conclusions may rely on certain assumptions).
4. The effectiveness to remove spurious correlations is impressing.

**Additional Comments:**

N/A

**Audience:**

Yes

**Audience Explanation:**

As what I stated above.

**Broader Impact Concerns:**

See in previous sections.

**Claims And Evidence:**

Yes

**Claims Explanation:**

The paper is generally well written, and the proposed method is supported by some mathematical analysis. The drawback is that the empirical evaluation is not very sufficient — the extracted concepts should be evaluated across a more diverse set of tasks.

**Requested Changes:**

1. More theoretical or empirical evidence of the correlations is needed: An important concern in this paper is why the authors can claim that the interpretable concepts or semantics are sensitive to rotation, unlike other noise patterns or artifacts. Although Section 3.1 provides some explanations, it is not sufficient to substantiate the paper’s most critical assumption.
(a) The paper implicitly assumes a one-to-one correspondence between concepts and directions. However, in real representation spaces, semantics often exhibit nonlinear or multidirectional distributions. For example, a concept may arise from a combination of multiple sub-directions or dimensions rather than a single vector direction. As a result, defining concepts purely through “directionality” may oversimplify the semantic structure.
(b) Furthermore, the paper only argues that “breaking rotational invariance” indicates the existence of some structure, but such structure may not necessarily be related to specific semantic concepts or particular tasks.
(c) The paper proposes replacing raw embeddings with singular vectors to mitigate scaling effects. However, linear decompositions such as SVD assume that the data follow linearly separable principal component structures; moreover, in high-dimensional neural representations, complex nonlinear relationships may not be fully captured through singular directions.

2. The experimental comparison provided in Sec.6.1 is not convincing enough, it seems that such vsualization is only provided with raw SVD, more comparisons with advanced methods are recommended.

---

> ### Author Response · Authors · 2026-01-25
> **Response to Reviewer MqaU**
>
> We thank the reviewer for recognizing our framework’s statistical rigor and our method's effectiveness in removing spurious correlations. Below, we address the reviewer's concerns in detail.
>
> * **On the linearity and polysemanticity of concepts**:
>     * For linearity, CLIP is trained using a contrastive loss that maximizes the dot product similarity between image and text embeddings. This objective explicitly enforces a linear structure on the final embedding space: if a concept (e.g., "dog") exists in the embedding space, the image embedding must have a high projection value onto the corresponding direction in text embedding space. Therefore, unlike intermediate layers, the semantic content in the final CLIP representation layer must be linearly decodable for the model to function. This justifies our use of singular vectors and linear decomposition (SVD/Varimax) as the correct tool for this specific space. We have added a clarification in Section 3.1 to explicitly discuss this "linear readout" property of Contrastive Learning.
>     * Regarding the concern that concepts may be entangled or multi-directional, we acknowledge that raw directions can be polysemantic. However, our method uses Varimax rotation specifically to minimize this entanglement. By maximizing the variance of squared loadings, we seek the simplest linear basis available. While this is a first-order approximation, our results on Waterbirds demonstrate that this linear disentanglement is sufficient to isolate and remove  spurious concepts like backgrounds.
>
> * **On the rotation sensitive structure and interpretability**: We agree that rotation sensitivity signals structure but does not by itself ensure interpretability. Our framework bridges this in two steps.
>     * First, our hypothesis testing framework uses rotation sensitivity to distinguish signal from noise. As shown in our random noise example, isotropic distributions are rotation-invariant (information is uniformly distributed). In contrast, rotation sensitivity reveals "preferred directions" in the embedding space, confirming the existence of a non-random, underlying data structure.
>     * Second, while not all such directions are inherently interpretable, our method uses Varimax rotation to specifically optimize kurtosis (sparsity), a standard objective for recovering meaningful and interpretable latent factors. Empirically, our spurious-correlation results (Sec. 6.3) show these directions are not random: removing learned concepts like background bias measurably changes model behavior.
>
> * **Clarifications of the comparison in Section 6.1**:
>     * We retained the comparison with raw SVD specifically to isolate the effect of rotation. Since SVD represents the unrotated basis, contrasting it with our method directly demonstrates that **the "natural" axes of the singular vectors are often polysemantic,** implying rotation is needed to align axes with human-interpretable concepts.
>     * To compare against a stronger, unsupervised baseline, we evaluated Independent Component Analysis (FastICA). Like our method, ICA seeks a linear transformation to disentangle signals, but it maximizes independence among concepts rather than sparsity.
>         * **Quantitative Result:** To quantify the "semantic purity" of the discovered concepts, we introduced a concept coherence score. For each concept direction, we retrieve the top-9 most activating images and calculate the average pairwise cosine similarity between their embeddings. A higher score indicates a tight, consistent semantic cluster, while a lower score indicates polysemantic concepts. As shown in Table 1, our method achieves the highest coherence, confirming it successfully minimizes concept entanglement.
>         * **Qualitative results:** On ViT-B-32 embeddings (50 concepts), FastICA recovered some  concepts (e.g., marine animals, dogs). However, it suffered from significant instability and semantic mixing in others (e.g., one concept mixed scorpions with boots).
>
> **Table 1: Quantitative Concept Coherence Scores**
> | Model | Weights | ICA Coherence | Raw SVD Coherence | Ours |
> | :--- | :--- | :--- | :--- | :--- |
> | ViT-B-32 | openai | 0.6490 | 0.6210 | **0.7644** |
> | RN50 | openai | 0.6128 | 0.6043 | **0.7435** |
> | ViT-L-14 | openai | 0.6427 | 0.6012 | **0.7251** |
> | ViT-B-32 | laion2b_s34b_b79k | 0.5784 | 0.4986 | **0.6765** |

---

### Review · Reviewer_Njv9 · 2026-01-11

**Summary Of Contributions:**

This paper proposes a statistically grounded framework for interpreting CLIP embeddings by explicitly testing whether the embedding space contains recoverable structure before attempting concept attribution. The paper appears more statistically rigorous than most concurrent papers aiming to attributing semantic concepts to embeddings, although a tradeoff might be lower benchmark performances.

**Audience:**

Yes

**Audience Explanation:**

people who work in vision and embedding will be interested to read the paper, i would also like to argue that everyone in the ML field should read it.

**Broader Impact Concerns:**

none noted.

**Claims And Evidence:**

Yes

**Claims Explanation:**

The paper's writting is fairly carful, not claims are well supported.

**Requested Changes:**

1. While the paper deliberately does not optimize for benchmark performance, including comparisons to stronger but potentially fragile baselines would help contextualize the tradeoffs. Even if performance is lower, such comparisons would clarify what is gained in terms of robustness and what is sacrificed.

2. Given that rotation sensitivity is central to the paper’s thesis, it would strengthen the claims to evaluate how competing methods behave under random orthogonal rotations of the embedding space. Demonstrating degradation in interpretability or performance for baselines would provide direct empirical support for the proposed framework.

3. I wonder if the authors can come up with some quantitative measures that are more reliable than the previous comparisons.

4. It seems the method is not strictly constrained to CLIP, some demonstration of its usage in other embeddings could be helpful.

---

> ### Author Response · Authors · 2026-01-25
> **Response to Reviewer Njv9 (1/2)**
>
> We thank the reviewer for their constructive feedback and recognizing our method's statistical rigor and writing. We address all the comments in details below.
>
> * **Comparisons to stronger but potentially fragile baselines (Response to request 1):** To address this, we have updated our evaluation with the following baselines:
>     * **Supervised linear probe (ERM):** We trained a logistic regression classifier on the Waterbirds training set using ground-truth labels. This represents a stronger supervised baseline that is directly optimized over prediction accuracy, whereas our method remains fully unsupervised.
>     * **Additional concept-based baselines:** We also expanded our comparison to include: non-negative matrix factorization (NMF), independent component analysis (ICA), Sparse Autoencoder (Sparse-AE) [1] and CRAFT (as a variant of NMF) [2]. All baselines use 50 concepts. Performance is measured by zero-shot classification accuracy after reconstructing embeddings with and without spurious components removed.
>     * **Results:** The results in Table 1 confirm the intuition regarding this trade-off. The ERM baseline achieves the highest average accuracy (94.9%) but is indeed fragile, collapsing to 29.2% on the worst-group. In contrast, our method sacrifices some average performance (89.6%) to gain substantial robustness (60.7%), verifying that we successfully identify and remove the spurious background concepts that the supervised baseline relies on.
>
> **Table 1: Spurious-Concept Removal Performance**
>
> | Method | Recon Avg | Recon WG | Spurious Removed Avg | Spurious Removed WG |
> | :--- | ---: | ---: | ---: | ---: |
> | ERM (Supervised) | — | — | **94.9%** | 29.2% |
> | NMF | 25.3% | 0.0% | 23.0% | 0.1% |
> | ICA | 86.5% | 40.8% | 80.6% | 52.1% |
> | CRAFT | 23.9% | 0.0% | 22.6% | 0.0% |
> | SVD | 85.5% | 39.0% | 70.7% | 60.4% |
> | Ours | 85.5% | 39.0% | 89.6% | **60.7%** |
>
> * **On the rotation sensitivity (Response to request 2):** To address this, we have added a new analysis. In this experiment, we applied random orthogonal rotations to the CLIP image embeddings and evaluated three methods: Raw SVD, SpLiCE, and Ours.
>     * **Ours:** Maintained high interpretability and low reconstruction error, confirming our framework successfully solves for the underlying structure even under arbitrary rotation.
>     * **SpLiCE:** Relies on projecting embeddings onto a fixed concept vocabulary, applying random rotations to the image space breaks the alignment between images and text embeddings. This resulted in a catastrophic increase in reconstruction error and a loss of semantic relevance, highlighting the fragility of fixed-concept methods.
>     * **Raw SVD** is robust to rotation transformations as singular vectors rotates with the data. The resulting concepts remained mixed. This again suggests variance-based decomposition is not enough, an explicit search for the "simple structure" rotation is required to disentangle the semantic concepts.
>
> * **On the quantitative results (Response to request 3):**
>     * We introduce a quantitative metric called concept coherence score to measure semantic purity of each concept. We calculate the average pairwise cosine similarity between the top-9 most activating images for each concept. A higher score indicates a tight semantic cluster. As shown in Table 2, our method achieves the highest coherence, confirming it minimizes concept entanglement better than ICA or Raw SVD.
>
> **Table 2: Quantitative Concept Coherence Scores**
> | Model | Weights | ICA Coherence | Raw SVD Coherence | Ours |
> | :--- | :--- | :--- | :--- | :--- |
> | ViT-B-32 | openai | 0.6490 | 0.6210 | **0.7644** |
> | RN50 | openai | 0.6128 | 0.6043 | **0.7435** |
> | ViT-L-14 | openai | 0.6427 | 0.6012 | **0.7251** |
> | ViT-B-32 | laion2b_s34b_b79k | 0.5784 | 0.4986 | **0.6765** |

---

> ### Author Response · Authors · 2026-01-25
> **Response to Reviewer Njv9 (2/2)**
>
> * **On other embedding models (Response to request 4):**
>     * Our framework is entirely model-agnostic and not strictly constrained to CLIP as it operates on the embedding space instead of model architecture. However, we initially focused on CLIP for two reasons:
>         * **Linear decodability:** CLIP is trained via a contrastive loss that maximizes the dot product between image and text embeddings. This objective explicitly enforces a linear structure: if a semantic concept exists, it must be linearly decodable in the shared embedding space. This makes CLIP ideal for verifying our rotation-based decomposition.
>         * **Automatic interpretation:** The aligned text encoder allows us to automatically interpret the discovered directions by projecting them onto text embeddings, rather than solely relying on human inspection of image cluster.
>     * To address the request for broader demonstration, we expand our evaluation to include models with different architectures and trained with different models, specifically we tested ResNet-50 (CNN-based) and OpenCLIP ViT-B-32 trained on LAION-2B.
>         * Across all tested models, we observed a consistent pattern: concepts derived from raw SVD (without rotation) frequently appeared as entangled mixtures. However, after applying our proposed rotation, the directions consistently aligned into semantically unified themes.
>         * When applied to the same dataset (ImageNet), different backbones (e.g., ResNet-50 vs. ViT) yielded similar semantic concepts to identify shared themes like dogs, knitwear, birds, and ships.
>         * These extracted concepts confirm that the rotation-sensitive structure is not an artifact of the ViT architecture. We have added these results to the appendix to demonstrate the method's broader applicability.
>
> **References:**
>
> [1] Cunningham et al. "Sparse autoencoders find highly interpretable features in language models." *arXiv* (2023).
>
> [2] Fel, T., A. Picard, L. Bethune, T. Boissin, D. Vigouroux, J. Colin, R. Cadène, and T. Serre. "CRAFT: Concept Recursive Activation FacTorization for Explainability." *Proceedings. IEEE Computer Society Conference on Computer Vision and Pattern Recognition* (2023): 2711–2721.

---

### Decision · Action_Editor_W8xT · 2026-02-09

**Recommendation:** Accept as is

**Additional Comments:**

Given that the authors did not submit a revised version of the paper but have provided several new results, it is expected that all of these will be incorporated into the paper and thoroughly checked during the camera-ready submission. For instance:

a) All tables should be added to the camera-ready version. In addition, the text should accommodate the new tables and expand the descriptions, concept explanation, and evaluations accordingly.

b) Similarly, for the random orthogonal rotations to the CLIP image embeddings related to rotation sensitivity

c) More explicitly report setup (how rotations are sampled/applied), what metrics are tracked (reconstruction error, interpretability proxies), and the qualitative/quantitative outcome that fixed-vocabulary alignment breaks while their approach remains stable.

d) All comments that provide additional explanations should be added to the right places in the manuscript.

**Audience:**

Yes

**Audience Explanation:**

The topic of this paper is very well aligned with TMLR's scope and audience, working on representation learning, interpretability, robustness, and multimodal models. A key reason is that it formalises a common but often implicit prerequisite in concept discovery pipelines, i.e. before interpreting discovered concepts, it is useful to test whether the representation actually contains non-isotropic structure that is recoverable beyond arbitrary rotations.

**Claims And Evidence:**

Yes

**Claims Explanation:**

This paper introduces an approach to statistically ground whether CLIP-style embedding spaces contain meaningful rotation-sensitive structure, and then extracts a sparse, more interpretable set of directions via simple structure rotation. The reviewers have been positive from the beginning, and the various queries were addressed appropriately with new experiments, a revised version, and related explanations.

One of the key concerns with these tests is simply detecting generic high-dimensional artefacts, but the revisions and explanations, along with newly added concepts, strengthen this argument.

On the interpretability side, the decomposition pipeline (SVD followed by a simple-structure rotation) is better argued for and motivated. The revised version will need to incorporate all these changes, hence offering more systematic evidence that the rotated factors are sparser and more semantically coherent, and that these directions can be acted on in downstream interventions. The spurious correlation experiments are especially useful here because they show that identified directions are not only nameable but also operationally meaningful, with measurable gains on worst-group performance after removing confounding concepts.